# Breaking the Limits of Autoregression! A Diffusion-Bridge with Mutual-Information for Time Series Forecasting

## Abstract

Time series forecasting (TSF) is a fundamental task in many real-world applications, yet effectively modeling both global dependencies and local dynamics remains challenging. Existing diffusion-based approaches typically adopt a noise-driven paradigm, which disrupts temporal continuity and fails to leverage intermediate evolutionary states, thereby limiting forecasting accuracy and robustness. To overcome these limitations, we propose **AR-DBMI** (**A**uto-**R**egressive **D**iffusion **B**ridge with **M**utual-**I**nformation correction), an autoregressive generative forecasting framework. AR-DBMI reformulates time series evolution as a "future-to-history" diffusion bridge, where intermediate states are deterministically generated through sliding operations to preserve transitional dynamics. To further enhance performance, we introduce a velocity-consistency constraint to capture first-order dynamics across windows, and a mutual-information alignment mechanism to ensure semantic consistency between predicted and ground-truth endpoints. In addition, dual-domain regularization combining time-domain anchoring and spectral consistency improves stability under non-stationary and noisy conditions. Extensive experiments conducted on seven widely used datasets demonstrate that our model achieves state-of-the-art performance, significantly outperforming existing diffusion-based TSF models.

## 1 Introdcution

Time series forecasting (TSF) (Box et al., 2015; Hamilton, 2020; Cheng et al., 2025) has been recognized as a critical task in a wide range of real-world domains, including finance (Li et al., 2024a; Black & Scholes, 1973; Li et al., 2024b), healthcare (Cheng et al., 2024), energy management (Li et al., 2025) and traffic planning (Long et al., 2024) of future data points can enable more informed decision-making, optimized resource allocation, and improved risk management, ultimately leading to substantial operational improvements and strategic advantages. With the rapid progress of deep learning, numerous advanced TSF models have been proposed, among which deep neural network–based approaches have emerged as a dominant and effective paradigm (Zhou et al., 2021; Lai et al., 2018; Wu et al., 2022). More recently, diffusion models have demonstrated remarkable potential in TSF tasks (Rasul et al., 2021; Fan et al., 2024; Yuan & Qiao, 2024).

Diffusion models, first popularized in computer vision, are characterized by a forward process that progressively injects noise and a reverse process that denoises to recover data (Ho et al., 2020; Jiang et al., 2025; Lu et al., 2023). While effective in vision, directly extending this noise-driven paradigm to TSF introduces fundamental mismatches. In recent years, several studies have introduced diffusion models into TSF tasks (Li et al., 2022; Shen & Kwok, 2023; Kollovieh et al., 2023). Time series are inherently continuous, yet conventional diffusion perturbs sequences into white noise, disrupting temporal continuity. More critically, intermediate evolutionary states—essential for modeling latent dynamics—are destroyed, preventing effective exploitation of temporal dependencies. These limitations lead to unstable performance, especially in non-stationary or noisy conditions. ARMD (Gao et al., 2025) attempted to mitigate this mismatch via sliding-based diffusion, preserving some transitional structure. However, its autoregressive formulation still restricts modeling to short-term dependencies, leaving long-horizon and complex dynamics insufficiently addressed.

To overcome these fundamental challenges, we propose **AR-DBMI** (**A**uto-**R**egressive **D**iffusion **B**ridge with **M**utual-**I**nformation correction), an autoregressive generative framework designed to align diffusion with the intrinsic temporal evolution of time series. The core idea lies in integrating the diffusion-bridge paradigm with mutual-information alignment to reformulate the "future-to-history" evolution of time series, thereby better aligning the diffusion mechanism with the intrinsic temporal dynamics and mitigating the short-term dependency limitations of autoregressive models. Specifically, time series segments are treated as the fundamental modeling units, where intermediate states are deterministically generated through sliding operations, thus preserving transitional information of temporal evolution. Furthermore, a diffusion-bridge velocity consistency constraint is introduced to enable the model to learn first-order dynamics between adjacent windows, ensuring continuity and directionality across time. In addition, we design an endpoint mutual-information embedding-level alignment mechanism that aligns predicted and ground-truth endpoints in the representation space through contrastive learning. To further enhance robustness, dual-domain consistency regularization is incorporated at the endpoint level, combining time-domain anchoring with spectral consistency to improve numerical stability and spectral fidelity. Finally, the effectiveness of the proposed method is evaluated on public datasets, demonstrating its superiority over existing competitive approaches. It is expected that the strong performance of AR-DBMI will inspire further research in this direction. The main contributions of this work can be summarized as follows:

- We propose AR-DBMI, an autoregressive diffusion-bridge framework that integrates deterministic sliding with mutual-information alignment to capture both global dependencies and local dynamics.

- We introduce a diffusion-bridge velocity consistency constraint and an endpoint-level mutual-information alignment module. These components significantly enhance the model's ability to capture first-order dynamics and align semantic representations, thereby improving both numerical accuracy and semantic fidelity.

- We validate AR-DBMI on seven widely used benchmark datasets and show that it delivers superior forecasting performance. The results consistently surpass existing diffusion-based models and establish state-of-the-art outcomes in time series forecasting.

## 2 RELATED WORK

**Time series forecasting.** In the early studies of TSF, most approaches relied on traditional statistical models such as ARMA and ARIMA (Box et al., 2015), which identified patterns under the assumption of linear relationships between past and present observations. More recently, deep learning models have gained prominence in TSF. Temporal Convolutional Networks (TCN) (Fan et al., 2024; Gao et al., 2023) employed convolutional layers to capture temporal dependencies, while recurrent neural networks demonstrated strong performance in modeling sequential data. Transformer-based models (Li et al., 2019) were also widely adopted for TSF. Notably, PatchTST (Nie et al., 2022) divided time series into multiple tokens and utilized attention mechanisms to learn their interrelationships. iTransformer (Liu et al., 2023) applied attention along the inverted dimension to capture multivariate correlations. In addition, DLinear (Zeng et al., 2023) demonstrated the effectiveness of linear models for TSF tasks.

**Diffusion-based TSF models.** In the domain of diffusion-based TSF models, TimeGrad (Rasul et al., 2021) was proposed to autoregressively generate future values, which was suitable for short-term forecasting but suffered from error accumulation and slow inference in long horizons. CSDI (Tashiro et al., 2021) mitigated autoregressive inference by incorporating a self-supervised strategy with input masking. SSSD (Lopez Alcaraz & Strodthoff, 2023) improved upon CSDI by addressing quadratic complexity through the use of structured state space models instead of Transformers, although both CSDI and SSSD were still affected by boundary inconsistencies (Lugmayr et al., 2022). TimeDiff (Shen & Kwok, 2023) alleviated these limitations by introducing inductive biases tailored for time series data. D3VAE (Li et al., 2022) employed a coupled diffusion probabilistic model for data augmentation, integrating multi-scale denoising score matching and disentangled multivariate latent variables to improve accuracy and stability. TSDiff (Kollovieh et al., 2023) adopted a self-guidance mechanism during inference, without altering the training procedure, and achieved strong performance in both forecasting and synthetic data generation. mrDiff (Shen et al., 2024) leveraged seasonal-trend decomposition for trend extraction combined with non-autoregressive

denoising. Similarly, MG-TSD (Fan et al.) utilized data granularity levels as intermediate diffusion targets to enhance predictive performance. Finally, Diffusion-TS (Yuan & Qiao) introduced an encoder–decoder Transformer framework with disentangled temporal representations, enabling the generation of high-quality multivariate time series samples from noisy inputs. ARMD (Gao et al., 2025) Inspired by the autoregressive moving average (ARMA) paradigm, ARMD introduced the first continuous sequential diffusion–based model for TSF.

## 3 METHODOLOGY

### 3.1 PROBLEM DEFINITION

Given a time series history window $X_{-L+1:0} \in \mathbb{R}^{L \times C}$, where $L$ denotes the look-back length and $C$ is the number of channels. The objective of time series forecasting (TSF) is to predict the future segment $X_{1:H} \in \mathbb{R}^{H \times C}$. In this paper we set $L = T = H$ to enable a continuous "future-to-history" evolution. During training, we form the concatenated sequence $S = [X_{-H+1:0}, X_{1:H}] \in \mathbb{R}^{2H \times C}$, and extract sliding windows of length $H$ from $S$ as the intermediate states of the diffusion process. The learning objective is to train a reverse (devolution) network that progressively maps intermediate windows closer to the historical segment back to a noise-free prediction of the future endpoint $\hat{X}_{1:H}$.

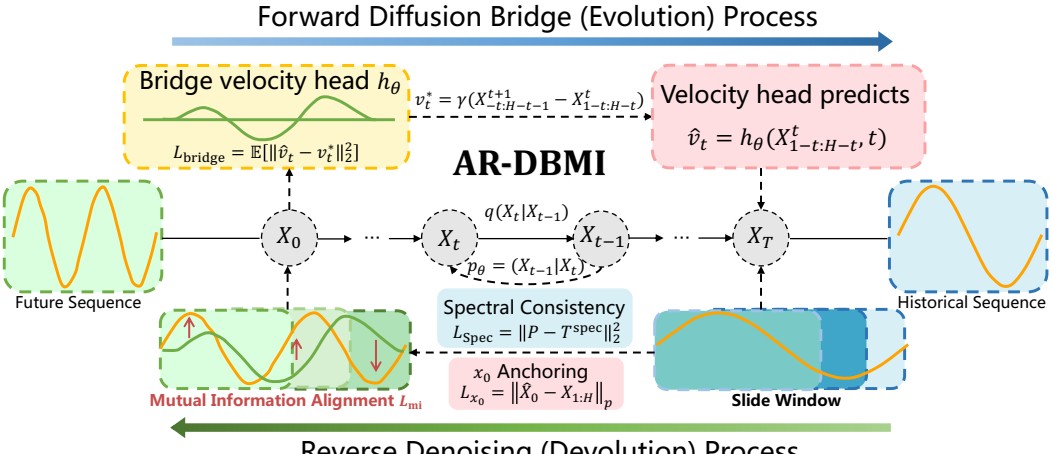

Figure 1: Illustration of the AR-DBMI framework. The future sequence $X^0_{1:H}$ evolves into the historical sequence $X^T_{-H+1:0}$ via deterministic sliding. A bridge velocity head enforces local temporal consistency, while a mutual information module aligns predicted futures with ground truth at the representation level.

### 3.2 THE PROPOSED AR-DBMI

This paper proposes AR-DBMI, an autoregressive time series model based on diffusion bridges, enhanced with mutual information correction. The overall framework of the model is illustrated in Fig. 1. AR-DBMI retains the auto-regressive moving design of ARMD and augments it with a Diffusion-Bridge head for velocity consistency and a Mutual-Information alignment loss. This model adopts the view of time series evolution as "future-as-initial, history-as-final". To avoid misunderstanding, we clarify that the term "autoregressive" in AR-DBMI refers to the iterative reverse-diffusion trajectory $(x_T \rightarrow x_{T-1} \rightarrow \cdots \rightarrow x_0)$, where each state is conditioned on the previous one in a DDIM-style deterministic manner. This is distinct from classical one-step-at-a-time AR forecasting, yet still constitutes a sequential, dependency-preserving generative process.

### 3.2.1 FORWARD DIFFUSION (EVOLUTION)

Unlike conventional diffusion-based time series models, AR-DBMI conceptualizes the evolution of a time series as a diffusion process. In this framework, the future sequence $X^0_{1:H}$ is regarded as the

initial state of the diffusion process, while the historical sequence $X^T_{-H+1:0}$ serves as the final state. Unlike traditional approaches that progressively inject noise, the intermediate states in AR-DBMI are generated deterministically through sliding operations. Specifically, let $\text{Slide}(X, k)$ denote shifting a window of length $H$ towards the historical direction by $k$ steps; then the one-step evolution can be expressed as:

$$X^t_{1-t:H-t} = \text{Slide}(X^{t-1}_{2-t:H-t+1}, 1), \qquad t = 1, \ldots, H-1. \tag{1}$$

We adopt the cumulative coefficient $\bar{\alpha}_t = \prod_{i=1}^{t} \alpha_i$, where $\alpha_i = 1 - \beta_i$. The window at an arbitrary step $t$ can then be expressed in closed form relative to the initial window:

$$X^t_{1-t:H-t} = \sqrt{\bar{\alpha}_t}\, X^0_{1:H} + \sqrt{1 - \bar{\alpha}_t}\, z^t. \tag{2}$$

Unlike noise injection in DDPM, the intermediate states in Eq. (2) are fully determined by the deterministic sliding operation. The residual term $z^t$ is not an injected noise variable but a closed-form expression that characterizes the deterministic difference between $X^0_{1:H}$ and $X^t_{1-t:H-t}$. It is introduced for notational convenience and serves as a supervisory signal during training. Specifically, $z^t$ represents the evolutionary trend of the series and can be computed as:

$$z^t = (\tfrac{1}{\sqrt{\bar{\alpha}_t}} X^t_{1-t:H-t} - X^0_{1:H})/\sqrt{\tfrac{1}{\bar{\alpha}_t} - 1}. \tag{3}$$

Under this scheme, the diffusion path is deterministic, and the maximum number of steps $H$ is set equal to the length of the target sequence to be predicted.

### 3.2.2 REVERSE DENOISING AND PARAMETERIZATION

At diffusion step $t$, the backbone network takes the intermediate window $x_t = X^t_{1-t:H-t}$ and predicts the clean endpoint $\hat{X}_0 = f_\theta(x_t, t)$. For training stability, we convert $\hat{X}_0$ into the corresponding residual form via the closed-form relation

$$\hat{\epsilon}_t = \frac{x_t - \sqrt{\bar{\alpha}_t}\hat{X}_0}{\sqrt{1 - \bar{\alpha}_t}}. \tag{4}$$

We then optimize an $\epsilon$-matching objective with SNR-aware weights. Given $\hat{X}_0$ and $\hat{\epsilon}_t$, the reverse mean under DDPM can be expressed as

$$\mu_\theta(x_t, t) = c_1(t)\hat{X}_0 + c_2(t)x_t, \tag{5}$$

where $c_1(t) = \frac{\beta_t \sqrt{\bar{\alpha}_{t-1}}}{1 - \bar{\alpha}_t}$ and $c_2(t) = \frac{1 - \bar{\alpha}_{t-1}}{1 - \bar{\alpha}_t}\sqrt{\alpha_t}$. During inference, we adopt the deterministic DDIM (Song et al., 2020) sampler with $\eta = 0$, ensuring that the reverse trajectory remains deterministic rather than stochastic:

$$x_{t-1} = \sqrt{\bar{\alpha}_{t-1}}\hat{X}_0 + \sqrt{1 - \bar{\alpha}_{t-1}}\hat{\epsilon}_t. \tag{6}$$

This deterministic update also supports skip-step rollbacks on sparse time grids, thereby accelerating sampling without introducing randomness. For completeness, $\epsilon$ and $v$ parameterizations and their bijective mappings to $(\hat{X}_0, \hat{\epsilon}_t)$ are summarized in Appendix C.2.

### 3.2.3 DIFFUSION-BRIDGE VELOCITY CONSISTENCY

To enable the model to explicitly learn the first-order dynamics (i.e., the local velocity field) of the "sliding evolution" between adjacent windows, we introduce an additional velocity head $h_\theta$ alongside the backbone. During training, this head takes the current intermediate window as input and predicts the velocity, i.e., the difference corresponding to a one-step slide toward the historical direction. The prediction is then aligned with the ground-truth velocity derived from the actual sliding operation, thereby incorporating both continuity and directionality across windows into the supervision. Throughout, we uniformly employ windows of length $H$ with $C$ channels.

$$v^*_t = \gamma\Big(X^{t+1}_{-t:,H-t-1} - X^t_{1-t:,H-t}\Big), \qquad \gamma > 0, \tag{7}$$

where $v^*_t$ denotes the target velocity at step $t$, and $\gamma$ is a scaling factor introduced to adjust the magnitude and dimensionality of the velocity term so that it remains balanced with the primary loss.

The window indices $[-t : H - t - 1]$ and $[1 - t : H - t]$ ensure that the two windows correspond to adjacent slides (a one-step shift toward the historical direction), such that their difference captures the evolutionary trend of a single-step transition.

Accordingly, the velocity head predicts

$$\hat{v}_t = h_\theta(X_{1-t: H-t}^t, t). \tag{8}$$

where $h_\theta$ shares the same time conditioning as the backbone (e.g., a timestep embedding) and outputs $v_t^* \in \mathbb{R}^{H \times C}$.

Although simple in form, the velocity head $h_\theta$ plays a specific and well-delimited role in our framework. In practice, $h_\theta$ is implemented as a single linear projection applied to the backbone features. This minimalist design is intentional: the purpose of the head is not to introduce additional representation depth, but rather to provide an explicit linear constraint that encourages the backbone to learn the first-order displacement structure between adjacent sliding windows. Such lightweight auxiliary heads are standard in diffusion-based and autoregressive representation-learning models, where linear mappings are commonly used to impose auxiliary supervision without interfering with the main generative pathway. The linear formulation also stabilizes optimization, as it yields a direct and well-conditioned gradient signal for the displacement field. In summary, the design of $h_\theta$ is specific to the role it plays in our model simple but effective and is aligned with established practices in related generative modeling frameworks.

To align the predicted velocity with the ground-truth velocity, we minimize the bridge-consistency loss

$$L_{\text{bridge}} = \mathbb{E}_t\left[\|\hat{v}_t - v_t^*\|_2^2\right], \tag{9}$$

the mean-squared error taken in expectation over timesteps $t$ sampled during training. In the overall objective this term is weighted by $\lambda_{\text{bridge}}$. Notably, the velocity head is used only during training to supply gradients that enforce local cross-window smoothness and directional consistency; it is not invoked at inference, so it incurs no additional sampling overhead.

### 3.2.4 MUTUAL-INFORMATION ALIGNMENT

To ensure that the model aligns with the ground-truth future segments not only at the numerical level but also in terms of semantic representation, we introduce a contrastive mutual information constraint between the endpoint outputs and their corresponding targets. Specifically, we first apply global average pooling along the temporal dimension to obtain a time-invariant global representation. This representation is then projected into a shared embedding space through a learnable low-dimensional projection. Using the InfoNCE objective, the predicted and ground-truth representations of the same sample are pulled closer together, while those of different samples are pushed apart, thereby achieving temporal calibration.

Given a batch $\left\{\hat{X}_0^i, \hat{X}_{1:H}^i\right\}_{i=1}^B$, where $\hat{X}_0^i$ is the model-predicted endpoint window, $\hat{X}_{1:H}^i$ is the ground-truth endpoint window.

We first perform temporal average pooling: $\text{mean}_h(\hat{X}_0^i) \in \mathbb{R}^C$, $\text{mean}_h(X_{1:H}^i) \in \mathbb{R}^C$. A learnable projection $\phi: \mathbb{R}^C \longrightarrow \mathbb{R}^d$ is then applied, followed by $\ell_2$-normalization, yielding the following query–key pairs: Query vector $q_i = \text{norm}(\phi(\text{mean}_h(\hat{X}_0^i))$ and key vector $k_i = \text{norm}(\phi(\text{mean}_h(X_{1:H}^{(i)}))$.

With a temperature parameter $\tau > 0$, the InfoNCE loss is defined as:

$$L_{\text{mi}} = \frac{1}{B}\sum_{i=1}^B\left[-\log\frac{\exp\left(q_i^\top k_i/\tau\right)}{\sum_{j=1}^B \exp\left(q_i^\top k_j/\tau\right)}\right]. \tag{10}$$

Here, $(q_i, k_i)$ form positive pairs within the same batch, while $(q_i, k_i)$, $j \neq i$ serve as negative pairs. The temperature $\tau$ controls the "sharpness" of the distribution: smaller $\tau$ values increase discriminability but may cause training instability, whereas larger $\tau$ values reduce sharpness and smooth the loss landscape. To mitigate numerical issues, we apply $\ell_2$-normalization so that cosine similarity effectively approximates residual correlation.

### 3.2.5 $x_0$ ANCHORING AND SPECTRAL CONSISTENCY

Beyond semantic-level alignment, we further impose time-domain and spectral-domain constraints directly on the predicted endpoint window to enhance stability and fidelity.

**Time-Domain Anchoring**: We apply an $\ell_p$ constraint directly on the endpoint window

$$L_{x_0} = \big\| \hat{X}_0 - X_{1:H} \big\|_p, \qquad p \in \{1, 2\}. \tag{11}$$

When $p = 1$, the loss is more robust to outliers and less sensitive to sharp errors; when $p = 2$ the loss is smoother and exhibits more stable gradients. In practice, this term is implemented consistently with the primary loss type and weighted by $\lambda_{x_0}$ in the total objective. This term provides the most direct numerical anchoring, which can substantially reduce endpoint drift and variance risks.

**Spectral Consistency**: To enforce consistency between the predicted sequence and the ground truth in terms of periodicity/frequency distribution, we apply a real-valued FFT along the temporal dimension. The resulting frequency dimension is $H_f = [H/2] + 1$, and we match the amplitude spectrum

$$L_{\text{spec}} = \big\| P - T^{\text{spec}} \big\|_2^2, \tag{12}$$

where $P = \big| \mathcal{F}(\hat{X}_0) \big|$ and $T^{\text{spec}} = \big| \mathcal{F}(X_{1:H}) \big|$. This amplitude-matching constraint aligns the energy distribution across frequencies, mitigating artifacts such as excessive high-frequency noise or oversmoothed low-frequency components. It is typically used together with time-domain anchoring: the former constrains structure and periodicity, while the latter ensures amplitude- and value-level fidelity. In implementation, this term is weighted by $\lambda_{\text{spec}}$ in the total objective.

---

**Algorithm 1** Training of AR-DBMI

**Input:** Max steps $T = H$; $(\alpha_t, \bar{\alpha}_t)$; SNR weights $w_t$; sampler SampleT; $\lambda_{\text{bridge}}, \lambda_{\text{mi}}, \lambda_{x_0}, \lambda_{\text{spec}}, \gamma, \tau$.
**while** not converged **do**
    Sample minibatch $S = [X_{-H+1:0}, X_{1:H}]$, form $X_{1:H}$
    $t \leftarrow \text{SampleT}(\{0, \dots, T-1\})$
    $x_t \leftarrow X_{1-t:H-t}^t, \hat{X}_0 \leftarrow f_\theta(x_t, t)$
    $\hat{\epsilon}_t \leftarrow \frac{x_t - \sqrt{\bar{\alpha}_t}\hat{X}_0}{\sqrt{1-\bar{\alpha}_t}}, \epsilon_t^* \leftarrow \frac{x_t - \sqrt{\bar{\alpha}_t}X_{1:H}}{\sqrt{1-\bar{\alpha}_t}}$
    $L_{\text{main}} \leftarrow \mathbb{E}[w_t \|\hat{\epsilon}_t - \epsilon_t^*\|_p]$
    $v_t^* \leftarrow \gamma(X^{t+1} - X^t); \ \hat{v}_t \leftarrow h_\theta(X^t, t)$;
    $L_{\text{bridge}} \leftarrow \|\hat{v}_t - v_t^*\|_2^2$
    Compute $L_{\text{mi}}$ (InfoNCE), $L_{x_0}, L_{\text{spec}}$
    $L \leftarrow L_{\text{main}} + \lambda_{\text{bridge}}L_{\text{bridge}} + \lambda_{\text{mi}}L_{\text{mi}} + \lambda_{x_0}L_{x_0} + \lambda_{\text{spec}}L_{\text{spec}}$
    $\theta \leftarrow \theta - \eta\nabla_\theta L$
**end while**
**Output:** trained parameters $\theta$

**Algorithm 2** Sampling / Forecasting of AR-DBMI

**Input:** $X_{-H+1:0}^T$; trained $f_\theta$; DDIM grid $\{t_0 = T > \cdots > t_S \geq 0\}$.
$x_{t_0} \leftarrow X_{-H+1:0}^T$
**for** $k = 0$ **to** $S - 1$ **do**
    $t \leftarrow t_k; \ t' \leftarrow t_{k+1}$
    $\hat{X}_0 \leftarrow f_\theta(x_t, t)$
    $\hat{\epsilon}_t \leftarrow \frac{x_t - \sqrt{\bar{\alpha}_t}\hat{X}_0}{\sqrt{1-\bar{\alpha}_t}}$
    **if** $t' < 0$ **then**
        $x_{t'} \leftarrow \hat{X}_0; $ **break**
    **else**
        $x_{t'} \leftarrow \sqrt{\bar{\alpha}_{t'}}\hat{X}_0 + \sqrt{1 - \bar{\alpha}_{t'}}\hat{\epsilon}_t$
    **end if**
**end for**
**Output:** $\hat{X}_{1:H} \equiv \hat{X}_0$

---

### 3.2.6 TRAINING OBJECTIVE AND SAMPLING

During training, we pair each sliding intermediate window $X_{1-t:H-t}$ with the future endpoint window $X_{1:H}$, thereby constructing a trend/noise target analogous to the noise prediction objective in DDPM:

$$\epsilon_t^* = \frac{X_{1-t:H-t}^t - \sqrt{\bar{\alpha}_t}\, X_{1:H}}{\sqrt{1 - \bar{\alpha}_t}}. \tag{13}$$

The primary loss is defined by mapping the model's output $\epsilon_t$ back to $\epsilon_t^*$, with weighting applied according to the signal-to-noise ratio (SNR):

$$L_{\text{main}} = \mathbb{E}_t \Big[ w_t \big\| \hat{\epsilon}_t - \epsilon_t^* \big\|_p \Big], \tag{14}$$

where $w_t = \kappa \frac{\sqrt{\alpha_t}\sqrt{1-\bar{\alpha}_t}}{\beta_t}, \kappa > 0$ is a scaling constant and $p \in \{1, 2\}$. The weight term $w_t$ emphasizes time steps with higher SNR, making training more stable and improving gradient utilization.

The final overall objective combines the main loss with auxiliary velocity and consistency terms:

$$L = L_{\text{main}} + \lambda_{\text{bridge}} L_{\text{bridge}} + \lambda_{\text{mi}} L_{\text{mi}} + \lambda_{x_0} L_{x_0} + \lambda_{\text{spec}} L_{\text{spec}}. \tag{15}$$

Here, each $\lambda$ controls the relative strength of its corresponding auxiliary component. In practice, a warm-up strategy can be adopted: auxiliary weights are initially set small so that the main loss dominates, and are then gradually increased to their target values, thereby achieving a better balance between stability and regularization. For better understanding, the training and sampling procedures are detailed in Algorithm 1 and Algorithm 2. Core code is detailed in Appendix D.3.

## 4 EXPERIMENTS

### 4.1 EXPERIMENTAL SETUP

**Datasets.** To evaluate AR-DBMI, we evaluate it on seven widely used time-series forecasting benchmarks: the four ETT datasets (ETTh1, ETTh2, ETTm1, and ETTm2), Solar Energy, Exchange, and Stock. The statistics of these datasets are summarized in Tab. 1.

**Baselines and Experimental Settings.** We compare AR-DBMI with five state-of-the-art (SOTA) diffusion-based TSF models: Diffusion-TS (Yuan & Qiao), MG-TSD (Fan et al.), TSDiff (Kollovieh et al., 2023), D3VAE (Li et al., 2022), TimeGrad (Rasul et al., 2021) and ARMD (Gao et al., 2025). We also include strong non-diffusion TSF baselines: iTransformer (Liu et al., 2023), TimesNet (Wu et al., 2022), DLinear (Zeng et al., 2023), PatchTST (Nie et al., 2022). For all datasets, the look-back length and the forecasting horizon are both fixed to 96. Following the evaluation protocol in (Li et al., 2025), we report mean squared error (MSE) and mean absolute error (MAE) computed on z-score–normalized data to ensure consistent comparison across variables. For all datasets, we construct training samples using overlapping sliding windows of total length 192, consisting of a 96-step look-back and a 96-step forecast horizon. The windows are shifted by one time step (k=1) between consecutive samples. The step size is defined on sequence indices rather than absolute time, so the same setting naturally applies to datasets with different sampling frequencies. Additional implementation details, including optimizer settings, loss-weight configurations, and diffusion-time parameters, are provided in Appendix C.

Table 1: The Statistics of Each Dataset.

| Dataset | Variables | Frequency | Length | Scope |
|---------|-----------|-----------|--------|-------|
| ETTh1/ETTh2 | 7 | 1 Hour | 17420 | Energy |
| ETTm1/ETTm2 | 7 | 15 Minutes | 69680 | Energy |
| Solar Energy | 137 | 1 Day | 52560 | Energy |
| Exchange | 8 | 1 Day | 7588 | Finance |
| Stock | 7 | 1 Day | 3685 | Finance |

### 4.2 COMPARISON WITH DIFFUSION-BASED MODELS

From Tab. 2, AR-DBMI achieves 12 out of 14 best results across seven datasets and two metrics. Relative to the second-best method, AR-DBMI further reduces MSE/MAE by 4.9%/3.5% on ETTh1; on ETTh2 it lowers MSE by 5.4% (with MAE only 1.8% higher, trading a lower squared error for a slightly less aggressive absolute error); on ETTm1 by 5.3%/1.9%; on ETTm2 by 2.8%/6.3%; and on Exchange by 6.5%/1.5%. On the most challenging Stock dataset—characterized by heavy noise and pronounced distribution shift—AR-DBMI reduces MSE/MAE by 77.9%/44.6% over the runner-up, demonstrating strong robustness and generalization. The only dataset where AR-DBMI trails ARMD is Solar Energy (MSE/MAE higher by 3.0%/2.5%); nevertheless, compared with Diffusion-TS it still achieves 5.0%/4.0% lower errors. Overall, AR-DBMI consistently delivers the lowest error in the vast majority of settings.

Table 2: Result comparisons of multivariate series forecasting with diffusion-based TSF models. The best results are highlighted in bold. The "Best Count" column indicates the times of achieving the best result.

| Methods | Metric | Solar Energy | ETTh1 | ETTh2 | ETTm1 | ETTm2 | Exchange | Stock | Best Count |
|---|---|---|---|---|---|---|---|---|---|
| **AR-DBMI** | MSE | 0.172 | **0.423** | **0.279** | **0.319** | **0.176** | **0.087** | **0.052** | **12** |
| | MAE | 0.242 | **0.443** | **0.344** | **0.369** | **0.239** | **0.200** | **0.149** | |
| ARMD | MSE | **0.167** | 0.445 | 0.311 | 0.337 | 0.181 | 0.093 | 0.235 | 1 |
| | MAE | **0.236** | 0.459 | 0.338 | 0.376 | 0.255 | 0.203 | 0.269 | |
| Diffusion-TS | MSE | 0.181 | 0.643 | 0.544 | 0.678 | 0.497 | 0.275 | 0.416 | 0 |
| | MAE | 0.252 | 0.586 | 0.494 | 0.613 | 0.459 | 0.382 | 0.533 | |
| MG-TSD | MSE | 0.443 | 1.096 | 0.295 | 0.690 | 0.202 | 0.396 | 0.365 | 0 |
| | MAE | 0.529 | 0.765 | 0.345 | 0.631 | 0.278 | 0.460 | 0.453 | |
| TSDiff | MSE | 0.352 | 0.614 | 0.470 | 0.686 | 0.242 | 0.125 | 0.330 | 0 |
| | MAE | 0.432 | 0.521 | 0.418 | 0.603 | 0.311 | 0.240 | 0.365 | |
| D3VAE | MSE | 0.416 | 1.123 | 0.389 | 0.644 | 0.394 | 0.240 | 0.345 | 0 |
| | MAE | 0.492 | 0.728 | 0.373 | 0.538 | 0.410 | 0.371 | 0.390 | |
| TimeGrad | MSE | 0.359 | 0.884 | 0.297 | 0.661 | 0.182 | 0.508 | 0.333 | 0 |
| | MAE | 0.449 | 0.725 | 0.349 | 0.639 | 0.254 | 0.554 | 0.376 | |

## 4.3 COMPARISON WITH OTHER TSF MODELS

As shown in Tab. 3, AR-DBMI achieves 9 of the 14 best results, substantially surpassing the other baseline models and demonstrating stable superiority across datasets and sampling frequencies. On the highly volatile Stock dataset, AR-DBMI reduces MSE/MAE from 0.286/0.325 to 0.052/0.149, evidencing strong robustness to heavy noise and distribution shift. It also delivers consistent gains on ETTh2, ETTm1, ETTm2, and Exchange. Although it is not the top performer on every dataset, AR-DBMI overall produces lower and smoother error curves than competing methods, combining accuracy, robustness, and transferability.

The overall advantage is evident in Fig. 2. In the diffusion-based comparison (panels a–b), AR-DBMI attains consistently lower and smoother MSE and MAE trajectories with markedly smaller cross-dataset fluctuations than competing diffusion models. In the comparison against mainstream non-diffusion TSF models (panels c–d), AR-DBMI likewise delivers the lowest or near-lowest errors, indicating stronger robustness and transferability. Mechanistically, the diffusion-bridge velocity consistency learns the first-order dynamics between adjacent windows; the mutual-information alignment draws the predicted endpoint toward the ground-truth endpoint in an embedding space; and the combination of $x_0$ anchoring and spectral consistency imposes inductive biases on time-domain amplitudes and frequency-domain energy. Together, these designs mitigate the uncertainty of pointwise denoising, producing a 2.8%–6.5% band of stable gains on the ETT family and Exchange, and 44.6%–77.9% improvements on the challenging Stock dataset. The "lower-is-better" trend of the line plots is consistent throughout, further supporting AR-DBMI's reliability under varying distributions and noise levels.

While AR-DBMI consistently improves over strong non-diffusion baselines, its advantage becomes most evident on datasets with high noise, distribution shifts, or multi-scale temporal variability. In such settings, deterministic models tend to underfit the intrinsic uncertainty, whereas diffusion-based generative models better capture stochastic dynamics and maintain stable long-horizon behavior. This explains the especially large gains observed on Stock and Exchange. On smoother, low-variance datasets, non-diffusion models already provide strong single-mode forecasts, leaving naturally smaller margins for diffusion-based methods.

## 4.4 EFFICIENCY COMPARISON

To assess the practical applicability of AR-DBMI, we further report the end-to-end training and inference time on the ETTm1 dataset. As shown in Table 4, AR-DBMI achieves substantially faster inference than existing diffusion-based baselines. This efficiency primarily results from the use of a single-step DDIM sampler and lightweight auxiliary heads, which introduce no iterative refinement

Table 3: Result comparisons of multivariate series forecasting with other TSF models. The best results are highlighted in bold.

| Methods | Metric | Solar Energy | ETTh1 | ETTh2 | ETTm1 | ETTm2 | Exchange | Stock | Best Count |
|---|---|---|---|---|---|---|---|---|---|
| **AR-DBMI** | MSE | **0.172** | 0.423 | **0.279** | **0.319** | **0.176** | 0.087 | **0.052** | **9** |
| | MAE | 0.242 | 0.443 | **0.344** | 0.369 | **0.239** | **0.200** | **0.149** | |
| iTransformer | MSE | 0.203 | 0.386 | 0.297 | 0.334 | 0.180 | **0.086** | 0.342 | 2 |
| | MAE | **0.237** | 0.405 | 0.349 | 0.368 | 0.264 | 0.206 | 0.413 | |
| TimesNet | MSE | 0.250 | **0.384** | 0.340 | 0.338 | 0.187 | 0.107 | 0.427 | 1 |
| | MAE | 0.292 | 0.402 | 0.347 | 0.375 | 0.267 | 0.234 | 0.499 | |
| DLinear | MSE | 0.290 | 0.386 | 0.333 | 0.345 | 0.193 | 0.088 | 0.286 | 1 |
| | MAE | 0.378 | **0.400** | 0.387 | 0.372 | 0.292 | 0.218 | 0.325 | |
| PatchTST | MSE | 0.234 | 0.414 | 0.302 | 0.329 | 0.175 | 0.088 | 0.516 | 1 |
| | MAE | 0.286 | 0.419 | 0.348 | **0.367** | 0.259 | 0.205 | 0.524 | |
| Client | MSE | 0.199 | 0.392 | 0.305 | 0.336 | 0.184 | 0.086 | 0.352 | 0 |
| | MAE | 0.239 | 0.409 | 0.353 | 0.369 | 0.267 | 0.206 | 0.433 | |

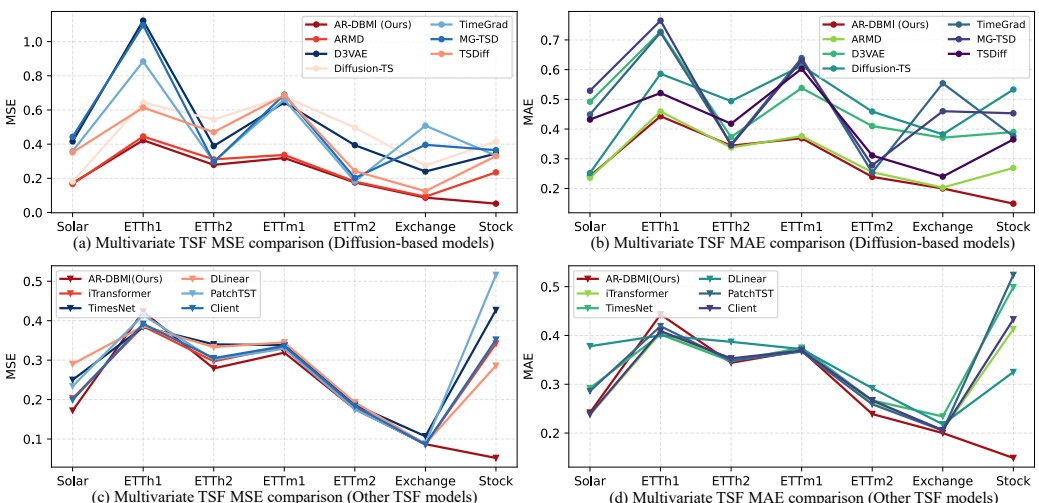

(a) Multivariate TSF MSE comparison (Diffusion-based models)

(b) Multivariate TSF MAE comparison (Diffusion-based models)

(c) Multivariate TSF MSE comparison (Other TSF models)

(d) Multivariate TSF MAE comparison (Other TSF models)

Figure 2: Cross-Dataset Error Curves (MSE/MAE, lower is better): AR-DBMI Shows the Lowest and Smoothest Trajectories.

or multi-stage decoding. Although AR-DBMI includes several auxiliary losses, all of them operate in $O(HC)$ and incur negligible computational overhead during both forward and backward passes.

Table 4: Training/inference time (s) of AR-DBMI and diffusion-based baselines on the ETTm1 dataset.

| Methods | Training time(s) | Inference time(s) |
|---|---|---|
| **AR-DBMI (Ours)** | **18.524** | **36.917** |
| Diffusion-TS | 149.974 | 1183.593 |
| MG-TSD | 834.815 | 3299.918 |
| TimeGrad | 449.792 | 1749.896 |

During training, AR-DBMI remains significantly more efficient than MG-TSD and TimeGrad, and is over 8× faster than Diffusion-TS. During inference, its runtime is more than 30× faster than Diffusion-TS, 90× faster than MG-TSD, and 45× faster than TimeGrad, demonstrating the practical advantage of the proposed design for real-world deployment.

## 4.5 ABLATION STUDY

We investigated the effectiveness of two key modules: the Mutual-Information (MI) alignment and Diffusion-Bridge velocity consistency. Using four configurations: the full AR-DBMI, *w/o MI*, *w/o Diffbri*, and *w/o MI-Diffbri*. The manipulations are as follows: **w/o MI**: We drop the temporal mean-pooling and learnable projection $\phi$ branch and set the InfoNCE loss $L_{\mathrm{mi}}$ to zero. Consequently, the backbone receives no gradients from semantic representation alignment and is trained only with numerical reconstruction–type objectives. **w/o Diffbri**: We remove the velocity head $h_\theta$ and its ground-truth target $v_t^*$, and set the bridge-consistency loss $L_{\mathrm{bridge}}$ to zero. The model therefore no longer learns the explicit first-order displacement/velocity between adjacent sliding windows; the cross-window continuity and directionality constraints are disabled, while all other training and sampling procedures remain unchanged. **w/o MI-Diffbri**: we remove the temporal mean-pooling and the learnable projection $\phi$ branch and set the InfoNCE loss $L_{\mathrm{mi}}$ to zero; meanwhile, we eliminate the velocity head $h_\theta$ and its ground-truth target $v_t^*$, and set the bridge-consistency loss $L_{\mathrm{bridge}}$ to zero. Consequently, the backbone receives neither semantic-alignment gradients nor explicit first-order displacement/velocity supervision between adjacent sliding windows; it is trained only with numerical reconstruction-type objectives together with endpoint/spectral consistency terms. All other training and sampling procedures remain unchanged. This configuration isolates the combined effect and tests the complementarity of the two modules.

Table 5: Comprehensive ablation study of AR-DBMI on ETTh2, ETTm2, and Stock. We examine the effect of removing: (i) Mutual-Information alignment (w/o *MI*), (ii) Diffusion-Bridge velocity consistency (w/o *Diffbri*), (iii) both (w/o *MI–Diffbri*), (iv) endpoint anchoring ($L_{x_0}$), (v) spectral consistency ($L_{\mathrm{spec}}$), and (vi) both endpoint-level consistency terms (w/o $L_{x_0}$–$L_{\mathrm{spec}}$).

| Dataset | AR-DBMI | | w/o *MI* | | w/o *Diffbri* | | w/o *MI–Diffbri* | | w/o $L_{x_0}$ | | w/o $L_{\mathrm{spec}}$ | | w/o $L_{x_0}$–$L_{\mathrm{spec}}$ | |
|---|---|---|---|---|---|---|---|---|---|---|---|---|---|---|
| | MSE | MAE | MSE | MAE | MSE | MAE | MSE | MAE | MSE | MAE | MSE | MAE | MSE | MAE |
| ETTh2 | **0.279** | **0.344** | 0.314 | 0.352 | 0.309 | 0.349 | 0.325 | 0.364 | 0.301 | 0.353 | 0.315 | 0.358 | 0.327 | 0.366 |
| ETTm2 | **0.176** | **0.239** | 0.192 | 0.261 | 0.196 | 0.273 | 0.203 | 0.285 | 0.189 | 0.263 | 0.198 | 0.276 | 0.207 | 0.289 |
| Stock | **0.052** | **0.149** | 0.212 | 0.271 | 0.227 | 0.273 | 0.242 | 0.289 | 0.204 | 0.262 | 0.238 | 0.283 | 0.257 | 0.299 |

To further isolate endpoint-level effects, we additionally remove: (i) the time-domain anchoring loss $L_{x_0}$, (ii) the spectral consistency loss $L_{\mathrm{spec}}$, and (iii) both. These ablations quantify the contributions of value-level anchoring and frequency-domain regularization. As shown in Table 5, AR-DBMI achieves the best performance across all metrics and datasets. Removing MI or Diffbri leads to moderate degradation, while removing both produces a more substantial drop. Eliminating $L_{x_0}$ causes endpoint drift, whereas removing $L_{\mathrm{spec}}$ introduces high-frequency artifacts—especially on the noisy Stock dataset. The joint removal of both consistency terms yields the worst performance among the endpoint-related variants.

Overall, the results demonstrate that all components contribute complementary benefits: MI alignment enhances semantic coherence, Diffbri enforces temporal smoothness, and $\{L_{x_0}, L_{\mathrm{spec}}\}$ stabilize the value and spectral structure. Together, these components enable AR-DBMI to remain robust across heterogeneous datasets and noise levels.

## 5 CONCLUSION

In this paper, we proposed AR-DBMI, an autoregressive diffusion TSF framework reframes time-series modeling as a future-to-history diffusion bridge rather than a noise-driven diffusion. Intermediate states are generated deterministically by sliding, preserving temporal continuity and making the reverse pass a natural de-evolution for forecasting. On this bridge, a velocity-consistency head learns first-order dynamics between adjacent windows, while a mutual-information alignment objective pulls the predicted and ground-truth endpoints in a shared embedding space. To enhance stability and fidelity, we add time-domain anchoring and spectral consistency, along with SNR-aware weighting and a noise-free DDIM update; the velocity head is train-only, adding no inference overhead. Across seven benchmarks, AR-DBMI yields lower and smoother errors and remains robust under noise and distribution shift, and ablations confirm the complementarity of velocity consistency and embedding-space alignment. This bridges diffusion mechanics with TSF objectives, jointly capturing global dependencies and local dynamics.

AUTHOR CONTRIBUTIONS

If you'd like to, you may include a section for author contributions as is done in many journals. This is optional and at the discretion of the authors.

ACKNOWLEDGMENTS

Use unnumbered third level headings for the acknowledgments. All acknowledgments, including those to funding agencies, go at the end of the paper.

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

REPRODUCIBILITY STATEMENT

To ensure the reproducibility of our results, we provide a comprehensive account of all experimental details. The full settings—including dataset preprocessing and hyperparameter configurations—are reported in the main text (see Sec. 4) and the appendix (see Appendix C). Clear pseudocode for the core algorithms is provided; see Algorithm 1 and Algorithm 2. The core of our PyTorch implementation is given in Appendix D.3. In addition, the complete implementation of AR-DBMI, together with scripts to reproduce all experiments, will be released upon publication.

ETHICS STATEMENT

This work constitutes foundational research in machine learning for time series forecasting. It does not involve human subjects, private data, or applications that pose direct ethical risks. Although the immediate societal impact of this theoretical study is limited, we acknowledge that future extensions—especially through our open-source codebase—may be applied across diverse domains. We recommend conducting domain-specific ethical reviews for any deployment in sensitive contexts to assess potential impacts.

## A  STATEMENT ON THE USE OF LARGE LANGUAGE MODELS (LLMs)

We used a large-language-model–based assistant only for language editing of the manuscript (grammar, wording, and stylistic consistency).

## B  DATASET DETAILS

We conducted extensive experiments on seven real-world datasets to evaluate the effectiveness of the proposed AR-DBMI across both diffusion-based and non-diffusion time-series forecasting (TSF) settings. The datasets span application scenarios with diverse periodic and trend characteristics, covering domains such as energy and finance. Detailed descriptions and the categorization of the datasets are provided in Tab. 1

**ETT** (Electricity Transformer Temperature): The ETT dataset encompasses two consecutive years of transformer oil temperature records along with six auxiliary electrical measurements collected from two counties in China. To facili- tate long-horizon forecasting at varying temporal granularities, we constructed four subsets: two with minute-level sampling (ETTm1 and ETTm2) and two with hourly sampling (ETTh1 and ETTh2). Each timestamp provides one temperature mea- surement accompanied by six associated features. Following standard practice, each subset is partitioned chronologically into 12 months of training data, 4 months for validation, and the remaining 4 months reserved for testing.

**Solar**: Solar-Energy records the power production of 137 photovoltaic plants in the state of Alabama (USA) during 2006, sampled every 10 minutes, for about 52,560 timestamps in total.

**Exchange**: The Exchange dataset contains eight major foreign exchange rates against the US Dollar, sampled daily from 1990 to 2016. A typical chronological split is 70%/10%/20% for train/validation/test.

**Stock**: The dataset does not have a single canonical public version in long-horizon forecasting benchmarks. In our study, it refers to a 7-variable daily financial time series with 3,685 timestamps.

## C  IMPLEMENTATION DETAILS

### C.1  EXPERIMENTS DETAILS

**Training Objectives and Loss Weights.**  We adopt an $\ell_1$ reconstruction loss as the main forecasting objective. Small auxiliary weights are applied to the additional objectives: bridge consistency ($\lambda_{\mathrm{bridge}} = 0.05$), mutual-information alignment ($\lambda_{\mathrm{mi}} = 0.01$, temperature $\tau = 0.1$, projection dimension 128), and endpoint anchoring ($\lambda_{x_0} = 0.02$). Spectral consistency is enabled or disabled depending on the experiment (see main text).

**Diffusion and Sampling Configuration.**  The number of diffusion timesteps is set to $T = 96$ to match the forecasting horizon. We employ a deterministic one-step DDIM sampler (`sampling_timesteps = 1`), so that the forward and reverse processes share the same temporal granularity. During training, time-step sampling follows the same DDIM scheme (`t_sample_mode = 'ddim'`).

**Optimization and Training Setup.**  Models are trained for up to 2300 epochs with Adam, an initial learning rate of $10^{-3}$, and a ReduceLROnPlateau scheduler with warm-up. The batch size is 128, and exponential moving averaging (EMA) is applied with decay 0.995.

**Data Preprocessing and Normalization.**  Datasets are split into train/test using an 80/20 proportion. All variables are normalized independently using z-score normalization. Sliding windows of total length 192 (96-step history + 96-step target) are generated with stride $k = 1$ as described in Section 4.1.

## C.2 ALTERNATIVE PARAMETERIZATIONS

**Main setting.** All primary experiments use the $x_0$-parameterization. For completeness, we list two alternative parameterizations and their bijective mappings to $(\hat{X}_0, \hat{\epsilon}_t)$.

**(i) $\epsilon$-parameterization.** Network output: $\hat{\epsilon}_t = f_\theta(x_t, t)$. The closed-form mapping is

$$\hat{X}_0 = \frac{x_t - \sqrt{1 - \bar{\alpha}_t}\,\hat{\epsilon}_t}{\sqrt{\bar{\alpha}_t}}, \qquad \hat{\epsilon}_t = \hat{\epsilon}_t.$$

**(ii) $v$-parameterization (rotated coordinates).** Define $v = \sqrt{\bar{\alpha}_t}\,\epsilon - \sqrt{1 - \bar{\alpha}_t}\,x_0$. If the network outputs $\hat{v}_t = f_\theta(x_t, t)$, the inverse maps are

$$\hat{X}_0 = \sqrt{\bar{\alpha}_t}\,x_t - \sqrt{1 - \bar{\alpha}_t}\,\hat{v}_t, \qquad \hat{\epsilon}_t = \sqrt{1 - \bar{\alpha}_t}\,x_t + \sqrt{\bar{\alpha}_t}\,\hat{v}_t.$$

*Note.* Once $(\hat{X}_0, \hat{\epsilon}_t)$ are recovered, the reverse mean under DDPM remains

$$\mu_\theta(x_t, t) = c_1(t)\,\hat{X}_0 + c_2(t)\,x_t, \quad c_1(t) = \frac{\beta_t \sqrt{\bar{\alpha}_{t-1}}}{1 - \bar{\alpha}_t}, \quad c_2(t) = \frac{1 - \bar{\alpha}_{t-1}}{1 - \bar{\alpha}_t}\sqrt{\alpha_t}.$$

## C.3 CHOICE OF PARAMETERIZATION IN OUR IMPLEMENTATION

We adopt the $x_0$-**parameterization** for all reported results: the backbone directly predicts $\hat{X}_0$, which is then converted to $\hat{\epsilon}_t$ via

$$\hat{\epsilon}_t = \frac{x_t - \sqrt{\bar{\alpha}_t}\,\hat{X}_0}{\sqrt{1 - \bar{\alpha}_t}}.$$

Training optimizes an $\epsilon$-**matching** objective with SNR-aware weights, while inference uses the deterministic **DDIM** sampler ($\eta = 0$). This combination is numerically stable and well-suited to one-/few-step sampling.

## C.4 TIMESTEP SELECTION AND TRAINING SCHEDULES

We support several timestep sampling policies during training:

- **uniform:** $t \sim \mathcal{U}\{0, \ldots, T - 1\}$;
- **snr:** $t$ is sampled proportionally to SNR-based weights;
- **tail:** softmax-biased toward large $t$ with a temperature-like factor;
- **ddim:** sample only the timesteps used at inference (e.g., when $S = 1$, $t = T - 1$).

In our runs, we typically use **snr** (default) or **ddim** for train–inference step alignment.

## C.5 AUXILIARY OBJECTIVES (TRAINING ONLY WHERE APPLICABLE)

**Bridge velocity consistency.** An auxiliary head predicts the one-step slide velocity with target

$$v_t^* = \gamma\left(X^{t+1} - X^t\right),$$

and the MSE loss

$$L_{\text{bridge}} = \mathbb{E}_t \left\| \hat{v}_t - v_t^* \right\|_2^2$$

enforces local continuity and directionality across adjacent windows. This head is used only during training and induces no inference overhead.

**Embedding-level alignment (InfoNCE).** Temporal mean pooling $\rightarrow$ projection $\phi : \mathbb{R}^C \rightarrow \mathbb{R}^d \rightarrow$ $\ell_2$-normalization, with temperature $\tau$:

$$L_{\text{mi}} = \frac{1}{B} \sum_{i=1}^{B} \left[ -\log \frac{\exp\left(q_i^\top k_i / \tau\right)}{\sum_{j=1}^{B} \exp\left(q_i^\top k_j / \tau\right)} \right].$$

**Endpoint anchoring and spectral consistency.** Time-domain anchoring

$$L_{x_0} \;=\; \left\| \hat{X}_0 - X_{1:H} \right\|_p, \qquad p \in \{1, 2\},$$

and frequency-domain amplitude matching

$$L_{\text{spec}} \;=\; \left\| \left| \mathcal{F}(\hat{X}_0) \right| - \left| \mathcal{F}(X_{1:H}) \right| \right\|_2^2.$$

All auxiliary terms are weighted in the total objective.

### C.6 SAMPLING DETAILS

We employ DDIM with $\eta = 0$ (deterministic). For fast sampling with $S$ steps, we follow the standard DDIM time grid; given $(\hat{X}_0, \hat{\epsilon}_t)$ at time $t$, the next update toward $t'$ is

$$x_{t'} \;=\; \sqrt{\bar{\alpha}_{t'}}\, \hat{X}_0 \;+\; \sqrt{1 - \bar{\alpha}_{t'}}\, \hat{\epsilon}_t,$$

where $t'$ is the next (earlier) timestep on the schedule. For numerical safety, we clip $\hat{X}_0$ when needed.

### C.7 THEORETICAL DISCUSSION ON STOCHASTIC VS. DETERMINISTIC DIFFUSION TRAJECTORIES

In classical diffusion probabilistic models such as DDPM, the forward process gradually corrupts data by adding Gaussian noise, forming a stochastic and Markovian latent trajectory $\{x_t\}_{t=1}^T$. The reverse process is then learned using score matching or noise prediction. Because the corruption is random, the resulting sampling trajectory naturally exhibits stochasticity.

A key theoretical finding of DDIM (Song et al., 2020) is that diffusion models also admit a *deterministic*, non-Markovian reverse process that yields samples from the same underlying data distribution as DDPM. In this formulation, the reverse dynamics follow a deterministic mapping

$$x_{t-1} = f_\theta(x_t, t),$$

where both stochastic and deterministic trajectories share the same fixed point $x_0$. DDIM shows that the deterministic process corresponds to selecting a specific solution of the probability-flow ODE governing diffusion models, differing only in the trajectory but not in the modeled data distribution.

Our formulation follows this deterministic-DDIM perspective. Instead of injecting Gaussian noise in the forward process, we employ a structured sliding operation as a deterministic corruption mechanism tailored to temporal evolution. Although the corruption differs from classical noise addition, the reverse process still learns to recover the clean endpoint $x_0$ via $x_0$-prediction, which is equivalent to the deterministic training objective used in DDIM. Thus, AR-DBMI operates as an implicit-likelihood diffusion model without requiring explicit likelihood estimation or score matching.

In summary:

- **Stochastic diffusion (DDPM)**: Gaussian noise in the forward process; stochastic, Markovian latent trajectories.
- **Deterministic diffusion (DDIM)**: noise-free, non-Markovian reverse process; trajectories differ but distribution remains identical.
- **AR-DBMI**: deterministic DDIM-style diffusion where sliding replaces noise as the corruption mechanism, preserving the reverse generative semantics while better matching temporal structure.

This establishes a principled link between AR-DBMI and the broader family of diffusion probabilistic models, situating our approach within the deterministic probability-flow interpretation while adapting the corruption mechanism to the structure of time-series forecasting.

## D TRAINING DETAILS

**Data preprocessing and windowing.** For all multivariate forecasting benchmarks, we follow the standard sliding-window protocol used in prior work. Each time series is first z-score normalized

per variable using the training split. We then construct overlapping windows of total length 192, consisting of a 96-step look-back and a 96-step forecast horizon. During training, windows are shifted by one time step, i.e., the stride between two consecutive samples is $k = 1$. The same configuration is used across all datasets to ensure a fair comparison.

**Backbone and optimization.** The AR-DBMI backbone is instantiated from the ARMD architecture with sequence length $H = 96$ and feature dimension $C$ matching the number of variables in each dataset. Unless otherwise specified, we use the cosine noise schedule with $T = 96$ diffusion steps and a single-step DDIM sampler ($S = 1$) at inference time. The model is trained with the $\ell_1$ reconstruction loss as the primary objective.

We adopt the Adam optimizer with a base learning rate of $1.0 \times 10^{-3}$, batch size 128, gradient accumulation of 2 steps, and a total of 2300 training epochs. An exponential moving average (EMA) of model parameters is maintained with decay 0.995 and update interval 10 steps. For the learning-rate schedule, we use a Reduce-on-Plateau strategy with warm-up: the learning rate is linearly increased from $8.0 \times 10^{-4}$ to $1.0 \times 10^{-3}$ over the first 500 optimization steps, and is subsequently reduced by a factor of 0.5 when the validation metric saturates, with a minimum of $1.0 \times 10^{-5}$.

**Auxiliary objectives.** AR-DBMI includes four auxiliary components on top of the main diffusion objective: (i) diffusion-bridge velocity consistency, (ii) mutual-information alignment (InfoNCE), (iii) time-domain endpoint anchoring $L_{x_0}$, and (iv) spectral consistency $L_{\mathrm{spec}}$. In our default configuration, the corresponding loss weights are

$$\lambda_{\mathrm{bridge}} = 0.05, \quad \lambda_{\mathrm{MI}} = 0.01, \quad \lambda_{x_0} = 0.02, \quad \lambda_{\mathrm{spec}} \in \{0, \text{small positive value in ablations}\},$$

where $\lambda_{\mathrm{spec}}$ is set to zero for the main benchmark tables and activated with a small positive value in spectral ablation experiments. The total training objective can be written as

$$\mathcal{L}_{\mathrm{total}} = \mathcal{L}_{\mathrm{main}} + \lambda_{\mathrm{MI}} \mathcal{L}_{\mathrm{MI}} + \lambda_{\mathrm{bridge}} \mathcal{L}_{\mathrm{bridge}} + \lambda_{x_0} \mathcal{L}_{x_0} + \lambda_{\mathrm{spec}} \mathcal{L}_{\mathrm{spec}}.$$

**Warm-up of auxiliary losses and learning rate.** To avoid destabilizing the early training phase, we apply a simple linear warm-up to both the learning rate and the auxiliary loss weights. Let $t \in \{1, \ldots, T_{\mathrm{train}}\}$ denote the current optimization step and $T_{\mathrm{w}}$ the warm-up length (we use $T_{\mathrm{w}} = 500$ steps in all experiments). For each auxiliary component $i \in \{\mathrm{MI}, \mathrm{bridge}, x_0, \mathrm{spec}\}$ with final weight $\lambda_i^{\mathrm{final}}$, the effective weight at step $t$ is

$$\lambda_i(t) = \begin{cases} \lambda_i^{\mathrm{final}} \cdot \dfrac{t}{T_{\mathrm{w}}}, & t < T_{\mathrm{w}}, \\ \lambda_i^{\mathrm{final}}, & t \geq T_{\mathrm{w}}. \end{cases}$$

Similarly, the learning rate is increased linearly from the warm-up value $\mathrm{lr}_{\mathrm{warmup}} = 8.0 \times 10^{-4}$ to the base value $\mathrm{lr}_{\mathrm{base}} = 1.0 \times 10^{-3}$ over the same warm-up period:

$$\mathrm{lr}(t) = \begin{cases} \mathrm{lr}_{\mathrm{warmup}} + \left(\mathrm{lr}_{\mathrm{base}} - \mathrm{lr}_{\mathrm{warmup}}\right) \dfrac{t}{T_{\mathrm{w}}}, & t < T_{\mathrm{w}}, \\ \mathrm{lr}_{\mathrm{base}}, & t \geq T_{\mathrm{w}}. \end{cases}$$

In practice, this schedule allows the backbone to first establish low-level predictive structure before the higher-level consistency and MI regularizers are fully activated, which we found to noticeably improve optimization stability and convergence.

### D.1 DETERMINISTIC NATURE AND STATISTICAL ROBUSTNESS OF AR-DBMI

AR-DBMI is designed as a fully deterministic diffusion–bridge model. In contrast to stochastic diffusion frameworks, AR-DBMI employs (i) a noise-free sliding operator in the forward process, (ii) a deterministic DDIM-style reverse process with $\eta = 0$, and (iii) fixed windowing and input normalization during inference. Consequently, both the evolution trajectory and the reverse denoising path are deterministic mappings without any injected randomness.

Because of this determinism, the variability across different training runs is extremely small. In our experiments, repeated training under different initialization seeds yields MSE/MAE differences

typically on the order of $10^{-4}$ to $10^{-5}$, which is well below the three-decimal precision commonly used in time-series forecasting benchmarks. As a result, seed-to-seed fluctuations become negligible after rounding, and reported metrics appear identical across multiple runs.

From an optimization perspective, the AR-DBMI backbone inherits the stable convergence behavior of ARMD—featuring fixed window structures, consistent gradient flows, and no stochastic sampling during training or inference. The EMA update and long training schedule further reduce the influence of initialization randomness, leading the model to converge reliably toward a consistent solution manifold.

Given that AR-DBMI exhibits near-zero seed variance and produces reproducible results across runs, the point estimates reported in the main tables already reflect the true performance of the method. While full statistical tables (e.g., mean ± std, confidence intervals, hypothesis tests) are standard practice in some domains, generating such statistics for every baseline would require retraining all diffusion-based models (e.g., MG-TSD, TimeGrad) across multiple seeds on seven datasets, which is computationally prohibitive.[1]

**Summary.** AR-DBMI's deterministic formulation naturally leads to high reproducibility and minimal seed-to-seed variation. Therefore, the point estimates in Tables 2–4 are stable, representative, and statistically robust even without multi-seed aggregation.

## D.2 SCALABILITY DISCUSSION

Although AR-DBMI is evaluated on seven widely used multivariate TSF benchmarks, these datasets share the common characteristics of mid-range sequence lengths, moderate dimensionality, and regular sampling frequency. For completeness, we provide a discussion of how AR-DBMI is expected to scale to more challenging real-world scenarios.

**Long-horizon forecasting** AR-DBMI adopts a deterministic DDIM-style reverse process without the accumulation of stochastic noise, and the diffusion-bridge supervision encourages consistent cross-window evolution. Both properties help mitigate error propagation in autoregressive decoding. Nevertheless, forecasting horizons substantially longer than those standardized in existing diffusion-based benchmarks (e.g., 192, 336, or 720 steps) may still introduce compounded uncertainty. Extending AR-DBMI to multi-scale or hierarchical temporal structures is a promising future direction for improving long-horizon stability.

**High-dimensional series** The auxiliary consistency terms (velocity, MI alignment, anchoring, spectral) act on window-level representations and are agnostic to the number of variables. The deterministic sliding mechanism also operates locally in time and does not depend on the dimensionality of the input. Therefore, AR-DBMI is expected to scale to higher-dimensional settings. However, extremely large variable sets (e.g., thousands of sensors) may require architectural adaptations such as tensorized linear layers, sparse attention, or low-rank parameterizations.

**Irregularly sampled or asynchronous series** The current formulation assumes uniformly sampled sequences. While the deterministic corruption mechanism can conceptually extend to irregular sampling by redefining the sliding operator on continuous-time embeddings, a full integration with event-based or asynchronous TSF models is an important direction for future work.

**Summary** AR-DBMI is designed with scalability in mind, and its architectural components generalize naturally beyond the mid-scale benchmarks used in the community. Nonetheless, long-horizon, high-dimensional, and irregular settings introduce additional modeling challenges that we consider valuable avenues for future extensions of the framework.

## D.3 CORE CODE

---

[1] For instance, MG-TSD requires approximately 3300 seconds for a single inference pass on ETTm1, and multi-seed training across all datasets would exceed feasible resource budgets.

**Algorithm 3** PyTorch implementation of AR-DBMI

**Listing 1** core of AR-DBMI

```
class ARDBMI(nn.Module):
    def __init__(self, seq_length, feature_size, pred_len=96, T=96,
                 mi_w=1e-2, tau=0.1, bridge_w=0.05, v_scale=1.0):
        super().__init__()
        self.pred_len, self.mi_w, self.tau = pred_len, mi_w, tau
        self.bridge_w, self.v_scale = bridge_w, v_scale
        self.f = Linear(n_feat=feature_size, n_channel=seq_length,
                       w_grad=True)
        self.h = Linear(n_feat=feature_size, n_channel=seq_length,
                       w_grad=True) if bridge_w > 0 else None
        self.proj = nn.Sequential(nn.Linear(feature_size,128), nn.GELU
                       (), nn.Linear(128,128)) if mi_w > 0 else None
        # cosine schedule -> buffers
        x = torch.linspace(0, T, T+1, dtype=torch.float64)
        a = torch.cos(((x/T)+0.008)/1.008 * math.pi * 0.5)**2; a = a /
            a[0]
        b = 1 - (a[1:]/a[:-1]); abar = torch.cumprod(1-b, dim=0)
        rb = lambda n,v: self.register_buffer(n, v.float())
        rb("abar", abar); rb("sa", abar.sqrt()); rb("soa", (1-abar).
        sqrt())
        rb("sra", (1/abar).sqrt()); rb("srm1", (1/abar-1).sqrt())
        rb("lw", ((1-b).sqrt()*(1-abar).sqrt()/b/100).float())
        rb("tprob", (torch.ones(pred_len-1)/(pred_len-1)).float())

    def _slide(self, x, t):
        i = torch.clamp(t+1, 1, self.pred_len-1)
        return x[:, self.pred_len - i : -i, :]

    def forward(self, x):
        B, dev = x.size(0), x.device
        t = torch.multinomial(self.tprob, B, replacement=True).to(dev)
        .long()
        xt = self._slide(x, t)
        y  = x[:, self.pred_len:, :]
        x0 = self.f(xt.clone(), t, training=True)
        eps = (self.sra.gather(0,t).view(B,1,1)*xt - x0) / self.srm1.
               gather(0,t).view(B,1,1)
        et  = (xt - y*self.sa.gather(0,t).view(B,1,1)) / self.soa.
               gather(0,t).view(B,1,1)
        main =(F.l1_loss(eps, et, reduction="none").view(B,-1).mean(1)
                * self.lw.gather(0,t)).mean()
        loss = main + F.l1_loss(x0, y)          # x0 anchoring

        if self.mi_w > 0 and self.proj is not None and B > 1:
            zp = F.normalize(self.proj(x0.mean(1)), dim=-1)
            zt = F.normalize(self.proj(y.mean(1)),  dim=-1)
            loss += self.mi_w * F.cross_entropy((zp @ zt.t())/self.tau
                       , torch.arange(B, device=dev))
        if self.bridge_w > 0 and self.h is not None:
            tc = torch.clamp(t, 0, self.pred_len-2)
            Xt  = self._slide(x, tc); Xt1 = self._slide(x, tc+1)
            loss += self.bridge_w * F.mse_loss(self.h(Xt.clone(), tc,
                       training=True),
            (Xt1 - Xt) * self.v_scale)
        return loss
```

