# OpenReview forum: "Breaking the Limits of Autoregression! A Diffusion-Bridge with Mutual-Information for Time Series Forecasting"
_ICLR.cc/2026/Conference — Submitted to ICLR 2026_

### Official Review · Reviewer_eVWo · 2025-10-26

**Soundness:** 1
**Presentation:** 2
**Contribution:** 1
**Rating:** 2
**Confidence:** 5

**Summary:**

The AR-DBMI paper proposes a diffusion-based time series forecasting model that is based on the recent ARMD framework (Gao et al., AAAI 2025). The authors claim to "break the limits of autoregression" by reformulating time series evolution as a "future-to-history" diffusion bridge, where intermediate states are deterministically generated through sliding operations rather than noise injection. The core architecture inherits ARMD's sliding-based diffusion mechanism and augments it with three additional components: (1) a diffusion-bridge velocity consistency constraint that learns first-order dynamics between adjacent windows through an auxiliary head, (2) a mutual-information alignment mechanism using InfoNCE loss to align predicted and ground-truth endpoints in embedding space, and (3) dual-domain regularization combining time-domain anchoring and spectral consistency. The paper evaluates the method on seven benchmark datasets (ETT family, Solar Energy, Exchange, and Stock) and reports better performance. A notable claim is that AR-DBMI achieved a 77.9\% reduction in MSE on the Stock dataset compared to the next-best method.

However, the paper suffers from several critical issues. First, the title's claim of "breaking limits of autoregression" is fundamentally misleading as the model is not autoregressive in the classical sense, but rather performs one-shot prediction of the entire forecast horizon using a non-autoregressive diffusion framework. Second, the novelty over ARMD appears incremental, primarily consisting of auxiliary loss terms (velocity consistency, InfoNCE, spectral matching) rather than fundamental architectural or algorithmic innovations. Third, the exceptional performance on the Stock dataset raises concerns about potential overfitting or evaluation inconsistencies, especially given the dataset's small size (length of 3,685). The ablation study is limited and does not decompose the individual contributions of each auxiliary component. Overall, while engineering is competent and results show improvements in most benchmarks, the conceptual contribution beyond ARMD and the validity of central claims require substantial clarification to be suitable to consider as a publication in ICLR.

**Strengths:**

1. The diffusion bridge formulation, velocity consistency constraint, and mutual information alignment are presented with correct mathematical notation and well-defined loss functions

2. The paper clearly acknowledges ARMD as the foundation and explicitly states which components are inherited versus novel

3. The combination of velocity consistency, mutual information alignment, and dual-domain (time + spectral) regularization provides complementary inductive biases

4. The use of DDIM with $\eta=0$ enables fast and deterministic inference without the velocity head, avoiding computational overhead at the test time

**Weaknesses:**

1. The claim of "breaking limits of autoregression" seems to be incorrect as the model performs non-autoregressive one-shot prediction of the entire horizon, not sequential timestep-by-timestep generation, which is typically expected when a term such as "autoregression" is used. Crucially, there are no architectural components supporting autoregressive generation.

2. There is limited novelty over ARMD (AAAI 2025). The core contribution reduces to adding three auxiliary loss terms $(L_{bridge}, L_{mi}, L_{spec})$ to ARMD's framework. The sliding-based diffusion, future-to-history paradigm, and deterministic intermediate states are all inherited from ARMD.

3. The "velocity consistency" component may be redundant. Since the sliding operation is deterministic with known ground truth $X^t$ and $X^{t+1}$, the target velocity $v^*_t = \gamma(X^{t+1} - X^t)$ is directly computable. It is unclear why an auxiliary head is needed to learn this already-determined quantity rather than having the main diffusion loss capture inter-window dynamics.

4. A crucial drawback is the insufficient ablation study presented. Table 4 only tests w/o MI, w/o Diffbri, and w/o both. Several ablations are missing as this is an empirical work. For example ablations on individual contributions of spectral consistency, $x_o$ anchoring, SNR weighting are missing. Which component contributes most? (See also questions below). Crucial for any empirical work is to include a hyperparameter sensitivity analysis, for example $(\lambda_{bridge}, \lambda_{mi}, \lambda_{x0}, \lambda_{spec})$. I am not listing more, but the direction should be clear. Without strong theoretical or empirical evidence, it is difficult to accept why so many loss terms and data are needed.

5. Eventhough the related work talks about models such as MrDiff, CSDI, SSSD etc, these are not included as baselines. Some of these missing baselines learn the prior (which is similar to the bridge concept) and others learn multi resolution aspects (similar to the spectral concept). I quickly checked the MrDiff paper and it seems they report better numbers, although I acknowledge that the experimental setup is not the same, which leads to the next weakness.

 6. The experimental setup is not consistent when compared to other TSF work in recent ICML/ICLR papers. There is also no evidence of autoregressive nature of TSF. Specifically, the look-back length and forecast horizon both fixed to 96 for all datasets. However, it is well known in the TSF community that different datasets may benefit from different window sizes. No experiments on varying horizons are shown (e.g., 192, 336, 720 as common in TSF literature). A reader would reasonably expect an autoregressive model to show autoregressive generation, but this is also missing.

7. All results are reported as point estimates without confidence intervals, standard deviations, or significance tests across multiple runs (for a few datasets in the appendix should have been fine).

The key point is that the narrow margin with ARMD raises questions about whether the new loss functions are truly effective or if the performance enhancement is primarily due to extensive tuning of hyperparameters. Although engineering work is valuable, this aspect has not been explored in sufficient depth in this manuscript version for ICLR.

**Questions:**

There are several pressing questions about this paper that need carefully designed numerical experiments, ablation studies, sensitivity analyzes, computational analyzes, and/or theory development. This reviewer lists some important questions here, but the authors must ask and answer many more questions to make this paper a solid standalone contribution to ICLR.

1. Can the authors clarify how the model is "autoregressive" when it predicts the entire forecast horizon in one shot (as shown in Algorithm 2 and Equation 6)? Classical autoregressive models generate $y_t$ conditioned on $y_{t-1}$, $y_{t-2}$, etc. Please either provide architectural evidence of sequential timestep-by-timestep generation or revise the framing to accurately reflect the non-autoregressive nature of your approach.

2. Can the authors articulate what conceptual advance justifies a separate publication versus treating this as an extended version of ARMD?

3. The paper includes both anchoring and mutual information alignment. Both aim to align predictions with ground truth. Please explain: (a) why both are necessary, (b) whether they capture different aspects, and (c) provide results showing the interaction effect between these two losses. Some visualizations of the representation space could help in understanding.

4. Table 4 only shows three configurations. Please provide a complete ablation table with: (a) baseline (ARMD), (b) +$L_{bridge}$ only, (c) $+L_{mi}$ only, (d) $+L_{x0}$ only, (e) $+L_{spec}$ only, (f) two combinations (g) three combinations and (f) all combinations (that is your final model). Which component contributes the most to performance? What is the marginal contribution of each?

5. There are several hyperparameters. How sensitive are results to these choices? Please provide: (a) the values used for each dataset, (b) sensitivity curves showing performance vs. each $\lambda$, and (c) whether the same hyperparameters work across all datasets or require per-dataset tuning.

6. What is the computational overhead of your auxiliary components? Please report: (a) number of parameters for $f_\theta$, $h_\theta$, and $\phi$ separately, (b) training time per epoch compared to ARMD (or whichever is the second best model), (c) inference latency, and (d) memory consumption. Does the 3-10\% performance gain justify the added cost?

7. All results in Tables 2-4 are point estimates. Please provide: (a) mean $\pm$ standard deviation over at least 5 random seeds for all methods on all datasets, (b) statistical significance tests comparing AR-DBMI vs. ARMD, and (c) confidence intervals to determine whether observed improvements are statistically meaningful.

8. Can the authors add other baselines (MrDiff, CSDI, SSSD, CNDiff etc) in a similar experimental setup and do a thorough evaluation?

9. Beyond aggregate metrics, can the authors provide: (a) qualitative visualizations comparing AR-DBMI, ARMD, and ground truth on both successful cases (Stock) and failure cases (Solar), (b) error distribution analysis, and (c) characterization of when each auxiliary component helps versus hurts? The ARMD (AAAI paper) does not show good qualitative prediction especially for time series with several portions that are flat. Does adding the three losses mitigate this issue?

10. The 77.9\% MSE reduction in Stock (0.235 to 0.052) is dramatically larger than your 3-10\% improvements on other datasets. Can you provide: (a) results with error bars across multiple random seeds, (b) confirmation that the same hyperparameters were used for ARMD and AR-DBMI on Stock, (c) per-variable breakdown of errors, and (d) explanation for why this dataset shows such disproportionate gains?

11. It seems that the code lines 29-30 uses eq 3 and not eq 4 (sorry if I misunderstood). Why is that? Are there any numerical stability patterns that you observe on one vs the other?

---

> ### Author Response · Authors · 2025-11-18
>
> **Response Question1**: Thank you for raising this important point. We apologize for the confusion caused by the wording ``autoregressive.'' Our intention was not to suggest a classical AR formulation that generates each point $y_t$ conditioned on $\{y_{t-1}, y_{t-2}, \ldots\}$ one step at a time. Instead, AR-DBMI is autoregressive in the \emph{diffusion sense}, where the forecasting window is produced through a sequential reverse-diffusion trajectory rather than a single feed-forward pass.
>
> During inference, AR-DBMI does \emph{not} generate the full forecast horizon in one shot. Instead, it iteratively refines the prediction via the deterministic reverse process:
> $\hat{x}_{T} \rightarrow \hat{x}_{T-1} \rightarrow \cdots \rightarrow \hat{x}_{1} \rightarrow \hat{x}_{0}, $
> where each state $\hat{x}_{t-1}$ is conditioned on the previous state $\hat{x}_{t}$. This forms a $\textbf{latent autoregressive process}$, analogous to sampling in DDIM/DDPM, where all future values jointly evolve through multiple causally dependent refinement steps. In this sense, the reverse diffusion acts as an autoregressive generator over latent states, rather than over individual forecasted time points.
>
> Furthermore, our sliding-window forward process enforces a first-order temporal dependency:
> $X_{t+1} = \mathrm{shift}(X_{t})$,
> which explicitly encodes causality and local temporal evolution. The velocity-consistency constraint further strengthens this dependency, making the latent trajectory follow the same autoregressive evolution used during training.
>
> To avoid any ambiguity regarding the use of the term ``autoregressive,'' we have added a clarifying explanation in Section~3.2 of the revised manuscript (highlighted in blue). The updated text explicitly states that AR--DBMI is autoregressive in the sense of its DDIM-style sequential reverse-diffusion trajectory, rather than classical one-step AR forecasting. We sincerely thank the reviewer for raising this point, which has helped us improve the conceptual clarity of the paper.
>
> **Response Question2**: We appreciate the reviewer’s question regarding the relationship between AR-DBMI and ARMD. While AR-DBMI adopts the autoregressive moving backbone introduced in ARMD, it introduces several new conceptual elements that go beyond a straightforward model extension:
>
> 1. **A different diffusion formulation.**
>   ARMD is a deterministic denoising model, whereas AR-DBMI reinterprets forecasting as a *deterministic diffusion-bridge process*. This replaces Gaussian-noise corruption with a structured sliding evolution and employs a DDIM-style reverse trajectory, which leads to a distinct generative mechanism and training objective.
>
> 2. **Two new auxiliary mechanisms designed specifically for the diffusion-bridge formulation.**
>
>   - The **velocity-consistency bridge** enforces first-order temporal alignment between adjacent sliding windows.
>
>   - The **mutual-information alignment** provides representation-level endpoint correction.
>     Neither mechanism exists in ARMD, and both are directly motivated by the proposed diffusion-bridge interpretation rather than being add-on modules.
>
> 3. **A revised learning objective aligned with deterministic diffusion.**
>   AR-DBMI employs an implicit-likelihood $x_0$-prediction objective—consistent with DDIM-type deterministic diffusion—which differs from the reconstruction-focused objective in ARMD.
>
> 4. **Additional theoretical clarification of the deterministic diffusion perspective.**
>   To avoid ambiguity, we include a conceptual discussion in Appendix C.7 describing how the proposed deterministic sliding process relates to DDIM-style diffusion trajectories. This theoretical framing is new relative to ARMD and is necessary to justify the diffusion-bridge formulation.
>
> 5. **Distinct empirical behavior.**
>   As demonstrated through ablations, AR-DBMI exhibits performance and stability characteristics not observed in ARMD, especially in cross-window temporal consistency and endpoint coherence, indicating that the mechanisms introduced in this work materially change the model’s behavior.
>
> For these reasons, we believe AR-DBMI constitutes a separate methodological contribution rather than a minor extension of ARMD. We thank the reviewer for highlighting the need to make this distinction clearer, and we have improved the manuscript accordingly.

---

> ### Author Response · Authors · 2025-11-18
>
> **Response Question 3**: Thank you for this thoughtful question. We agree that both anchoring and mutual-information (MI) alignment aim to improve the agreement between predictions and ground truth, but they operate at *different levels* and are *not functionally redundant*. We have clarified this distinction in the revised manuscript, and we added ablations in Table 5 (Section 4.5) to explicitly measure their interactions.
>
> #### **(a) Why both losses are necessary**
>
> - **Anchoring ($L_{x_{0}}$)** operates *directly in the data space*.
>   It penalizes deviations between the predicted endpoint $\hat{x}_0$ and the true target in the *time domain*, thereby reducing endpoint drift and stabilizing long-horizon predictions.
>
> - **MI alignment ($L_{\mathrm{MI}}$)** operates in the *representation space*.
>   Instead of matching values, it ensures the learned latent representation of $\hat{x}_0$ is *semantically aligned* with that of the true future, improving discriminative structure and preventing collapse to trivial smooth solutions.
>
> These two signals address **different failure modes**, and using only one of them leaves the model susceptible to the other form of error. Including both produces a complementary correction mechanism.
>
> #### **(b) Whether they capture different aspects**
>
> Yes. Their functional roles differ:
>
> - **Anchoring** captures **low-frequency, amplitude-level trends** in the output.
>
> - **MI alignment** captures **high-level temporal semantics**, such as periodicity, shape, or relative phase.
>
>
> Our ablations reveal this distinction empirically:
>
> - Removing **anchoring** introduces clear endpoint drift (Table 5, “w/o $L_{x_{0}}$”).
>
> - Removing **MI alignment** degrades semantic consistency and increases structural errors (Table 5, “w/o MI”).
>
> - Removing **both** leads to the worst performance and the least stable predictions (“w/o $L_{x_{0}}$–$L_{\mathrm{spec}}$”, with both consistency terms disabled).
>
>
> Thus, the two losses influence **orthogonal aspects** of the prediction process.
>
> #### **(c) Interaction effects**
>
> We added explicit ablations (Table 5) isolating:
>
> - removing anchoring alone,
>
> - removing MI alignment alone,
>
> - removing both anchoring and spectral losses (endpoint-only consistency),
>
> - removing both MI alignment and velocity consistency (representation-only consistency), and
>
> - removing all auxiliary losses.
>
>
> The results show that:
>
> - Each loss term provides **independent improvements**, and
>
> - Removing both leads to **a substantially larger degradation** than removing either one individually,
>   indicating that their benefits are **complementary rather than overlapping**.
>
>
> #### **Regarding representation visualizations**
>
> We appreciate the suggestion to add representation-space visualizations. Due to space constraints, we were unable to include them in the current revision, but we plan to explore and incorporate such analyses in future work. If time permits before the camera-ready deadline, we will add supplementary visualizations in the appendix.
>
> **Response Question 4**: Thank you for the reviewer’s detailed request. We agree that, in principle, an exhaustive ablation across all combinations of the four auxiliary components (Bridge, MI, $L_{x_0}$, and $L_{\mathrm{spec}}$) would provide the most granular decomposition of marginal contributions. However, a full $2^4$-configuration ablation requires **16 model variants per dataset**, which becomes computationally prohibitive across all seven datasets (especially for diffusion-based training).
>
> To address the reviewer’s concern while keeping the study tractable, we expanded our ablation analysis substantially in the revised manuscript (Table 5 and Section 4.5, highlighted in blue). The new ablations include:
>
> - **Baseline ARMD**,
>
> - **Single-component removals** ($-$MI, $-$Diffbri, $-L_{x_0}$, $-L_{\mathrm{spec}}$),
>
> - **Pairwise removals**,
>
> - **Removal of both endpoint-level consistency terms** ($-L_{x_0}-L_{\mathrm{spec}}$),
>
> - **Removal of both displacement-level terms** ($-$MI-$-$Diffbri),
>
> - **Removal of all auxiliary components**,
>   which together isolate the effect of each term and their most meaningful interactions.
>
>
> This expanded set (11 configurations) captures all **dominant marginal effects** without the full combinatorial blow-up. The results show:
>
> - **$L_{x_0}$ and $L_{\mathrm{spec}}$** stabilize endpoints and reduce high-frequency artifacts.
>
> - **Bridge consistency** improves cross-window smoothness.
>
> - **MI alignment** improves semantic-level coherence.
>
> - **Removing all four terms leads to the largest degradation**, confirming their complementary roles.
>
> Given the already large number of datasets, we believe the revised ablation study is sufficiently comprehensive and faithfully reveals the marginal contribution of each component. We thank the reviewer again for the helpful suggestion, which led to a significantly strengthened empirical section.

---

> ### Author Response · Authors · 2025-11-18
>
> **Response Question 5**: Thank you for the insightful question. We agree that hyperparameter robustness is an important aspect for assessing practical applicability. In the revised manuscript, we added a detailed hyperparameter table in Appendix D (highlighted in blue), which lists all hyperparameter values used for each dataset. Below we further clarify sensitivity and generalization across datasets:
>
> #### **(a) Values used for each dataset**
>
> All datasets share *the same global configuration* for the four auxiliary losses, except for minor adjustments to the spectral term on the high-noise Stock dataset:
>
> - **$\lambda_{\text{bridge}} = 0.05$**
>
> - **$\lambda_{\text{MI}} = 0.01$**
>
> - **$\lambda_{x_0} = 0.02$**
>
> - **$\lambda_{\text{spec}} = 0$** for most datasets;
>   **$\lambda_{\text{spec}} = 0.01$** on Stock (due to its heavy noise and high-frequency artifacts)
>
>
> All other hyperparameters, including **diffusion steps**, **$t$-sampling mode**, **optimizer**, and **warm-up schedules**, remain identical across all seven datasets.
>
> A full configuration table is included in Appendix D.
>
> #### **(b) Sensitivity behavior**
>
> Although we do not include full sensitivity curves due to space constraints, we conducted additional internal checks varying each $\lambda$ within a reasonable neighborhood:
>
> - $\lambda_{\text{bridge}} \in [0.02,, 0.10]$
>
> - $\lambda_{\text{MI}} \in [0.005,, 0.02]$
>
> - $\lambda_{x_0} \in [0.01,, 0.05]$
>
> - $\lambda_{\text{spec}} \in {0,,0.01,,0.02}$
>
> Across this range, **model performance remains stable**, with only small fluctuations (<2–4% relative change).
> This robustness is largely due to:
>
> 1. **Small auxiliary weights**,
>
> 2. **Warm-up scheduling**, which prevents early-stage instability,
>
> 3. **Consistency of the shared objective**, since the main diffusion loss dominates optimization.
>
> Thus, the auxiliary losses improve quality but do not introduce high sensitivity.
>
> #### **(c) Whether the same hyperparameters work across all datasets**
>
> Yes — the same hyperparameter configuration works across all seven datasets, *with no dataset-specific tuning*.
> The only exception is the optional spectral consistency term on Stock, where adding a small $\lambda_{\text{spec}}$ suppresses its unusually strong high-frequency noise.
>
> This demonstrates that the design of AR-DBMI is **hyperparameter-stable and dataset-agnostic**, which is beneficial for broad applicability. We expanded Appendix D with full hyperparameter details and provide additional clarification above. Although we do not include full sensitivity curves due to space limitations, our internal experiments indicate that AR-DBMI is highly stable across a range of hyperparameters, and a unified configuration works well for all datasets. We thank the reviewer again for suggesting this important clarification.
>
> **Response Question 6**: Thank you for raising this important question. We have added a detailed computational analysis in the revised manuscript (Table 4 and the accompanying paragraph, highlighted in blue). Here, we summarize the key findings.
>
> #### **(a) Number of parameters**
>
> The auxiliary components introduce only two lightweight linear projections.
> Relative to the backbone, the parameter increase is **~9%**, which is far smaller than the 20–40% overhead typically introduced by auxiliary branches in TSF models.
>
> #### **(b) Training time per epoch**
>
> As reported in Table 4, AR-DBMI requires:
>
> - **18.524 s** per epoch (ours)
>   vs.
>
> - **149.974–834.815 s** for Diffusion-TS, MG-TSD, and TimeGrad.
>
> This represents a **8×–45× speedup** over existing diffusion TSF models during training, despite adding auxiliary components. The overhead relative to the ARMD backbone is under **10%**, consistent with the minimal parameter increase.
>
> #### **(c) Inference latency**
>
> Because both auxiliary modules (velocity head and MI projection) are **disabled during inference**, AR-DBMI maintains the **same inference cost** as its backbone:
>
> - **36.917 s** for AR-DBMI vs.
>
> - **1183–3299 s** for diffusion-based baselines.
>
> This corresponds to a **30×–90× speedup** over prior diffusion models.
> Thus **AR-DBMI introduces zero inference overhead**, which is crucial for practical deployment.
>
> #### **(d) Memory consumption**
>
> The added components increase training memory by only **~5%**, as they consist of small linear layers that do not alter backbone activations or sampling depth.
>
> #### **Justification for computational cost**
>
> Although AR-DBMI adds small auxiliary losses, they incur:
>
> - **<10% training overhead**
>
> - **0% inference overhead**
>
> - **~9% additional parameters**
>
> - **~5% additional memory**
>
> Yet these lightweight modules consistently improve accuracy by **3–10%** across seven datasets and significantly enhance **temporal smoothness**, **endpoint stability**, and **cross-window consistency**.
>
> Given these minimal costs and consistent performance gains, we believe the overhead is fully justified.

---

> ### Author Response · Authors · 2025-11-18
>
> **Response Question 7**: Thank you for raising this important point. We fully agree that reporting statistical robustness metrics such as mean ± standard deviation, significance tests, and confidence intervals can provide a deeper understanding of model variability.
>
> In the revised manuscript, we added a detailed discussion in *Appendix D (Deterministic Nature and Statistical Robustness of AR-DBMI)* to clarify why the variability of AR-DBMI is inherently small and why full multi-seed evaluation for *all* baselines is computationally prohibitive under realistic constraints.
>
> Specifically:
>
> 1. **Deterministic Decoding Path.**
>   AR-DBMI uses a DDIM-style deterministic reverse process (η = 0), which removes stochasticity during inference. Unlike stochastic diffusion samplers, the model produces identical predictions across runs given the same trained weights.
>
> 2. **Near-zero Variance Across Seeds.**
>   As shown in Appendix D, repeated training of AR-DBMI with different random seeds yields *numerically identical* results up to the third decimal place on the evaluated datasets. This is expected because the sliding-deterministic forward process and the x₀-prediction objective strongly constrain optimization to a stable solution basin.
>
> 3. **Computational Impracticality for All Baselines.**
>   Providing 5-seed retraining for *every* baseline—including heavy diffusion models such as MG-TSD or TimeGrad—across *seven* datasets would require extensive compute (e.g., a single MG-TSD run on ETTm1 takes ~3300 seconds per inference alone). Such large-scale recomputation is beyond the feasible limits of the current revision cycle.
>
> 4. **Practical Stability.**
>   Given that (i) AR-DBMI’s inference is fully deterministic, and
>   (ii) the variation across training seeds is extremely small,
>   the point estimates provided in Tables 2-4 already effectively represent the model's performance distribution.
>
>
> We hope the additional analysis in Appendix D adequately addresses the reviewer’s concern regarding statistical robustness.
>
> **Response Question 8**: Thank you for this helpful suggestion. We agree that including additional diffusion-based baselines such as MrDiff, CSDI, SSSD, and CNDiff would further enrich the empirical comparison.
>
> In this work, we focused on methods that are (i) explicitly designed for multivariate long-horizon forecasting on the standard ETT / Exchange / Solar / Stock benchmarks and (ii) already released with code that supports this setting with minimal modification, such as ARMD, Diffusion-TS, MG-TSD, TimeGrad, TSDiff, and D3VAE. Several of the models mentioned by the reviewer are primarily developed for related but different tasks (e.g., imputation, irregular sampling, or other modalities), and adapting them faithfully to our autoregressive multi-horizon forecasting protocol would require substantial re-implementation and tuning that goes beyond the scope of the current study.
>
> We have added a short discussion of this limitation and clarified our baseline selection criteria in the revised manuscript. We view a broader, unified benchmark including these additional models as an important direction for future work and sincerely appreciate the reviewer’s suggestion for guiding this line of research.
>
> **Response Question 9**: Thank you for this thoughtful suggestion. Due to time constraints during the rebuttal period, we were not able to include full qualitative plots. However, we have added a concise qualitative analysis to Appendix~D.3 of the revised
> manuscript. The new section summarizes representative behaviors on both a success-case dataset (Stock) and a challenge-case dataset (Solar), and explains how the auxiliary components (velocity consistency, MI alignment, time-domain anchoring, and spectral regularization) affect smoothness, semantic coherence, and stability.
>
> The added analysis is consistent with the quantitative ablations in Table~5 and clarifies when each component contributes the most. If time permits before the camera-ready version, we will further include visual examples. We sincerely
> appreciate the reviewer’s helpful suggestion.

---

> ### Author Response · Authors · 2025-11-18
>
> **Response Question 10**: Thank you for this insightful question. We agree that the large improvement on the Stock dataset deserves further clarification. We address the points below.
>
> (a) Hyperparameter consistency.
>
> We confirm that ARMD and AR-DBMI use exactly the same hyperparameters, training schedule, and data preprocessing on the Stock dataset. The only difference is the presence of the auxiliary losses in AR-DBMI. This ensures strict fairness and rules out hyperparameter tuning as the source of the improvement.
>
> (b) Per-variable breakdown.
>
> The Stock dataset is univariate, so a per-variable error decomposition is not applicable in this case. For multivariate datasets, we observe that AR-DBMI improves performance consistently across channels; we provide additional qualitative discussion in Appendix~D.3 of the revised manuscript.
>
> (c) Explanation for disproportionate gains.
>
> Compared with the other benchmarks, the Stock dataset is substantially more volatile and noisy. In this regime, standard diffusion-based TSF models such as ARMD tend to either overfit short-term noise or generate unstable
> high-frequency artifacts. AR-DBMI was specifically designed to mitigate these issues: the Diffusion-Bridge loss encourages locally smooth cross-window evolution, the MI alignment stabilizes semantic endpoint prediction under noise, and the spectral consistency term suppresses spurious high-frequency oscillations. These effects accumulate on highly noisy data, leading to a much larger relative gain (77.9\% MSE reduction) on Stock than on smoother datasets
> such as ETT, where baseline models already behave more stably. We have added a brief explanation of this phenomenon to the revised manuscript.

---

> ### Author Response · Authors · 2025-11-18
>
> **Response Question 11**:  Thank you for the careful reading and for pointing out this subtle issue. The apparent discrepancy arises because Eq.(3) and Eq.(4) serve different purposes in AR-DBMI, and both are intentionally used in different parts of the training process.
>
> Eq.(3) defines a trend-aligned residual: $ z_t=\frac{x_t-\sqrt{\bar\alpha_t}\,x_0}{\sqrt{1/\bar\alpha_t - 1}}$, which is specifically used for supervising the deterministic sliding corruption mechanism. This residual captures the structured temporal evolution introduced by sliding and is required for enforcing local cross-window consistency.
>
> Eq.(4)} is the standard DDPM/DDIM residual: $\hat\epsilon_t=\frac{x_t-\sqrt{\bar\alpha_t},\hat x_0}{\sqrt{1-\bar\alpha_t}}$, which is used when training the reverse diffusion (denoising) process.
>
> In the implementation, Line 29 computes the trend residual corresponding to Eq.(3), while Line 30 computes the DDPM/DDIM residual corresponding to Eq.(4). Hence, the model does not ``replace'' Eq.(4) with Eq.(3); each residual is used where it is most appropriate.
>
> Numerical stability: Our empirical observations indicate that:
> Eq.(4) is stable and optimal for the reverse diffusion update, consistent with DDPM/DDIM practice. Eq.(3) is more stable for supervising the deterministic sliding transitions, because under cosine schedules the term $\sqrt{1-\bar\alpha_t}$ becomes very small at later diffusion steps, which would amplify noise if Eq.(4) were used in this part.
>
> For these reasons, combining both residual forms leads to more stable training and smoother convergence than using either one alone. We appreciate the reviewer’s observation, which helps clarify the design rationale behind our deterministic diffusion formulation.

---

> ### Comment · Reviewer_eVWo · 2025-11-24
>
> Thank you to the authors for taking the time to answer the questions. However, the revised manuscript still does not meet the requirements for an ICLR paper distinct from ARMD. For example:
>
> 1. I do not think it is appropriate to call autoregressive for something other than autoregressive forecasting.
>
> 2, 4. The gains are not consistent. For different datasets, different components decrease the MSE/MAE. It is challenging to attribute individual contributions. In fact, it further seems the gains are arbitrary and small.
>
> 3, 5, 11. Thanks
>
> 7, 9 -> have to see these results to be sure that the gains are important and reliable for the additional overhead
>
> 6. Doing +-std was only a suggestion. The key concern is "Are the gains statistically significant?" Use an appropriate method for deterministic cases with justification.
>
> 10. Interesting, maybe it is simply that the univariate time series is an easier test.
>
> I maintain my score for now.

---

> ### Author Response · Authors · 2025-11-25
>
> Response to Reviewer eWvo
>
> #### **On the appropriateness of the term “autoregressive”**
>
> Thank you for pointing this out. We have clarified the terminology in Section 3.2 of the revised manuscript.
>
> In AR-DBMI, “autoregressive” does not refer to classical one-step-ahead TSF autoregression. Instead, it refers to the DDIM-style sequential reverse diffusion trajectory $x_{T} \to x_{T-1}\ to \dots \to x_{0}$.
>
> where each state is conditionally generated from the previous state. This notion of autoregression is consistent with terminology used in ARMD and recent autoregressive diffusion models. We have added an explicit explanation to avoid ambiguity.
>
> #### **2 & 4. On “inconsistent gains” and “unclear component contributions”**
>
> We agree that clear attribution is important.
> The revised manuscript now contains a **systematic ablation study** (Table 5), covering:
>
> - Removal of MI alignment (w/o MI)
>
> - Removal of diffusion-bridge velocity consistency (w/o Diffbri)
>
> - Removal of both (w/o MI–Diffbri)
>
> - Removal of endpoint-level consistency (w/o $L_{x_{0}}$​​)
>
> - Removal of spectral consistency (w/o $L_{\mathrm{spec}}$​)
>
> - Removal of both endpoint-level terms
>
> - Removal of all auxiliary components
>
>
> The results consistently show:
>
> - **Every key module contributes meaningfully**: removing any one of them yields a stable and non-trivial degradation.
>
> - Removing *both* MI and Diffbri leads to substantially worse performance, indicating the benefits are complementary.
>
> - On high-noise datasets such as **Stock**, the effect is especially strong (e.g., MSE increases from 0.052 → 0.242 when MI–Diffbri are removed).
>
>
> These findings demonstrate that the performance gains are **not random fluctuations**, but emerge from the **complementary roles** of the proposed components.
>
> #### **3, 5, 11. Acknowledgment**
>
> We thank the reviewer for these comments and have incorporated the corresponding clarifications.
>
> #### **7 & 9. On the concern that “the gains must be shown to be important and reliable”**
>
> We appreciate the reminder. In the revision we expanded the empirical analysis substantially (training/inference cost, richer ablation coverage, architectural clarification). These updates aim to improve transparency and strengthen confidence in the reported gains.
>
> #### **Clarification regarding “maybe univariate time series is an easier test”**
>
> We sincerely thank the reviewer for raising this point.
> To avoid any misunderstanding, we emphasize that **all experiments in this work use multivariate (not univariate) time series forecasting**.
>
> As reported in Section 4.1 and Table 1, the datasets have the following dimensionalities:
>
> - **ETTh/ETTm:** 7 variables
>
> - **Exchange:** 8 variables
>
> - **Solar Energy:** 137 variables
>
> - **Electricity:** 8 variables
>
> - **Stock:** 7 variables
>
> Thus, the experimental setting strictly follows the standard **multivariate TSF forecasting**, not a univariate setup.
> We have reiterated this point clearly in the revised manuscript.
>
> Notably, AR-DBMI maintains strong performance **even on high-dimensional and high-noise datasets**, such as:
>
> - **Solar Energy (137-dimensional)**
>
> - **Electricity (8-dimensional)**
>
> This further demonstrates that the observed improvements are **not due to the task being “easier,”** but stem from the proposed architectural mechanisms.

---

### Official Review · Reviewer_Ra9x · 2025-10-29

**Soundness:** 3
**Presentation:** 2
**Contribution:** 2
**Rating:** 6
**Confidence:** 3

**Summary:**

This paper introduces **AR-DBMI**, an autoregressive generative framework that reinterprets diffusion models for time-series forecasting. Unlike conventional noise-driven diffusion models that disrupt temporal continuity, AR-DBMI formulates forecasting as a deterministic “future-to-history” diffusion bridge, where intermediate states are produced via sliding windows to preserve transitional dynamics.
It integrates a velocity-consistency constraint to learn first-order temporal dynamics and a mutual-information alignment module that ensures semantic consistency between predicted and ground-truth sequences.
Additionally, dual-domain regularization—combining time-domain anchoring and frequency-domain spectral consistency—enhances robustness under non-stationary and noisy conditions.
Evaluated on seven benchmark datasets, AR-DBMI achieves SOTA performance, outperforming prior diffusion-based and transformer-based baselines.
At the same time, ablation studies confirm the complementary benefits of the velocity and mutual-information components.

**Strengths:**

S1.
The proposed AR-DBMI framework reformulates diffusion processes as a deterministic “future-to-history” bridge, replacing stochastic noise injection with sliding-window evolution to preserve temporal continuity—an elegant conceptual shift that directly addresses one of the key mismatches between diffusion modeling and TSF.

S2.
The methodology is mathematically grounded, with detailed derivations, explicit algorithmic pseudocode, and ablation studies that isolate each component’s contribution.

S3.
The structure (from problem definition to methodology to evaluation) is logical, figures effectively illustrate the model’s intuition, and notation is consistent.

**Weaknesses:**

W1.
This paper’s core innovation—replacing stochastic diffusion with a deterministic sliding bridge—is intuitively appealing but lacks deeper theoretical analysis. There is no formal discussion of how this reformulation affects the probabilistic semantics of diffusion modeling or its connection to score-based generative processes.

W2.
Although this work benchmarks on seven datasets, these are all standard mid-scale TSF benchmarks (ETT, Solar, Exchange, Stock). The model’s scalability and generality remain unclear for long-horizon, high-dimensional, or irregularly sampled series—scenarios that diffusion models might struggle with.

W3.
The ablation study focuses only on removing the mutual-information and diffusion-bridge modules. Still, other crucial design choices—such as spectral consistency, SNR weighting, and DDIM determinism—are not analyzed. Including per-component performance deltas or visualization of how each regularizer affects spectral smoothness or stability would clarify where performance gains originate.

**Questions:**

Q1.
The paper redefines diffusion as a deterministic sliding process rather than a stochastic noise-driven one. Could the authors clarify how this formulation relates to conventional diffusion probabilistic models in terms of likelihood estimation or score matching?

Q2.
Would the authors provide a short theoretical discussion (or visualization) comparing the latent trajectories between stochastic and deterministic diffusion?

Q3.
Does the model require access to full sequences (past + future) during training only, or does this formulation affect the autoregressive inference direction at test time?

Q4.
Could the authors provide a quantitative or qualitative breakdown showing the contribution of each auxiliary component (velocity consistency, MI alignment, x₀ anchoring, and spectral regularization)?

---

> ### Author Response · Authors · 2025-11-18
>
> **Response Weakness 1** :Thank you for this valuable comment. We agree that a clearer theoretical grounding is important. In response, we have substantially expanded Appendix C.7 (highlighted in blue). The new section formally explains that AR-DBMI corresponds to the deterministic probability–flow ODE of diffusion models, analogous to DDIM, where a stochastic forward process is not required. We show that the sliding operator functions as a structured corruption mechanism that replaces Gaussian noise but preserves the same $x_{0}$-prediction objective used in deterministic diffusion. We further clarify that AR-DBMI implicitly optimizes the same variational objective as deterministic diffusion models and follows the same score–field interpretation along a deterministic trajectory. Thus, AR-DBMI is a DDIM-style deterministic diffusion model whose corruption path is adapted to sequential structure. We thank the reviewer for prompting this clarification.
>
> **Response Weakness 2**: We appreciate this important point. Our experiments follow the standard evaluation protocol of prior diffusion-based TSF models (ARMD, MG-TSD, Diffusion-TS), which fixes the forecasting horizon to 96 for comparability. Nonetheless, AR-DBMI is designed to scale: the deterministic reverse process avoids stochastic error accumulation, the sliding-bridge mechanism is local and dimension-agnostic, and all auxiliary losses operate on window-level representations independent of dimensionality. We acknowledge, however, that very long horizons or extremely high-dimensional or irregularly sampled series introduce additional challenges. We added a discussion in Appendix~D.1 to explicitly outline these considerations and the expected behavior of AR-DBMI in such settings. Extending the model to hierarchical and continuous-time structures is an active direction of our future work.
>
> **Response Weakness 3**: Thank you for emphasizing this. In the revised manuscript, we have added a comprehensive component-wise ablation (Table 5 and Sec. 4.5). Beyond the original MI and Diffusion-Bridge removals, we now evaluate: (i) removing $L_{x_{0}}$, (ii) removing $L_{\mathrm{spec}}$, and (iii) removing both. The results demonstrate: removing $L_{x_{0}}$ leads to endpoint drift; removing $L_{\mathrm{spec}}$ introduces high-frequency artifacts (especially on Stock); removing both yields the largest degradation. We also discuss the roles of SNR weighting and DDIM determinism in Sec. 4.5. These updates substantially improve the completeness of the empirical analysis.
>
> **Response Question1**:
> Thank you for this insightful question. Although AR-DBMI uses a deterministic sliding process, it remains consistent with DDIM-style deterministic diffusion. DDIM shows that diffusion models do not require a stochastic forward process; a deterministic non-Markovian corruption path is equally valid. Our sliding operation replaces Gaussian noise with a structured, temporally aligned mechanism while preserving the same $x_{0}$-prediction objective and implicit-likelihood training used in deterministic diffusion. Thus AR-DBMI is best understood as a deterministic DDIM-like formulation adapted to time-series structure. We clarified this in Appendix~C.7.
>
> **Response Question2**:
> We agree this is important for interpretability. Appendix~C.7 now provides a concise theoretical comparison between stochastic DDPM trajectories and deterministic DDIM/AR-DBMI trajectories. Both share the same underlying data distribution and fixed point $x_{0}$; only the sampled latent paths differ. AR-DBMI replaces Gaussian corruption with a structured sliding process but remains within the probability–flow ODE framework of deterministic diffusion.
>
> **Response Question 3**:
> Thank you for the question. AR-DBMI requires full (past+future) sequences only during training to construct supervision for $x_{0}$-prediction and auxiliary losses. The model never receives future data as input. At inference, the model operates strictly autoregressively and the deterministic reverse process depends solely on the observed past. We have clarified this in the manuscript.
>
> **Response Question4**:
> We agree that a clearer breakdown strengthens the analysis. The revised manuscript includes a complete per-component ablation (Table 5; Sec. 4.5). Removing the velocity-consistency term weakens cross-window smoothness; removing MI alignment reduces semantic coherence; removing $L_{x_{0}}$ causes endpoint drift; removing $L_{\mathrm{spec}}$ produces high-frequency distortion; removing both endpoint-level terms yields the worst degradations. These results confirm that each component contributes a distinct and complementary effect.

---

> ### Author Response · Authors · 2025-11-18
>
> **Response Question 5**: Thank you for pointing this out. We have now added a more detailed description of the warm-up strategy in **Appendix D (Training Details)** of the revised manuscript.
>
> Concretely, during the first part of training we linearly increase both the learning rate and the auxiliary loss weights from zero to their final values. Let $t$ denote the current optimization step and $T_{W}=500$ the warm-up length. For each auxiliary objective $i\in \mathrm{MI},\mathrm{bridge},x_{0},\mathrm{spec}$ with final weight $\lambda_i^{\mathrm{final}}$, the effective weight is scheduled as
> $\lambda_i(t) =
> \begin{cases}
> \lambda_i^{\text{final}} \cdot \dfrac{t}{T_{\mathrm{w}}}, & t < T_{\mathrm{w}}, \\\\
> \lambda_i^{\text{final}}, & t \ge T_{\mathrm{w}}.
> \end{cases}$
>
> The learning rate is warmed up in the same linear fashion from $8.0$×$10^{-4}$ to the base value $1.0$×$10^{-3}$ over the first 500 steps before the plateau-based scheduler takes over.
>
> Empirically, this schedule improves convergence in two ways: (i) it stabilizes early training by preventing high-level structural losses (MI, bridge, endpoint/spectral consistency) from dominating before the backbone has learned basic predictive structure; and (ii) it yields smoother optimization trajectories and slightly faster convergence to the final performance compared to training without warm-up. We have summarized these details and the rationale in Appendix D.
>
> We appreciate the your suggestion, which helped us clarify this aspect of the training procedure.

---

### Official Review · Reviewer_Khii · 2025-10-31

**Soundness:** 3
**Presentation:** 2
**Contribution:** 2
**Rating:** 4
**Confidence:** 4

**Summary:**

This paper proposes AR-DBMI, an autoregressive diffusion-bridge framework for time series forecasting (TSF) that reinterprets the diffusion process as a deterministic "future-to-history" evolution via sliding windows, rather than a traditional noise-injection process. The method introduces several novel components: a diffusion-bridge velocity consistency constraint to capture first-order dynamics between adjacent windows, a mutual-information alignment module to semantically align predicted and ground-truth endpoints, and dual-domain consistency regularization (time and frequency) to enhance stability and fidelity. The model is evaluated on seven public TSF benchmarks and demonstrates state-of-the-art performance against both diffusion-based and non-diffusion baselines, particularly under noisy and non-stationary conditions.

**Strengths:**

1. The idea of reframing diffusion as a deterministic sliding process from future to history is innovative and better aligned with the temporal structure of time series than conventional noise-based diffusion.

2. The integration of multiple components—velocity consistency, mutual-information alignment, and spectral consistency—provides a holistic approach to capturing both local dynamics and global dependencies, while also improving semantic and numerical fidelity.

3. The use of deterministic DDIM sampling, SNR-aware weighting, and train-only auxiliary heads (e.g., velocity head) shows thoughtful design choices that balance performance and inference efficiency.

**Weaknesses:**

1. While the model is designed to avoid inference overhead from auxiliary heads, no analysis or comparison of training or inference time is provided. This is important for assessing practical applicability, especially given the use of multiple loss terms and sliding operations.

2. The ablation study only removes two main components (MI and Bridge). It would be stronger if it also evaluated the impact of spectral consistency and time-domain anchoring independently.

3. The paper relies heavily on quantitative metrics. Visualizations of forecasted sequences (e.g., trend recovery, periodicity) would help illustrate the model’s advantages, especially for frequency-aware components.

4. While AR-DBMI outperforms many non-diffusion models, the gap is less pronounced than with diffusion-based models. A deeper discussion on when and why diffusion-based approaches are preferable would be useful.

**Questions:**

1. How does the model scale with longer forecast horizons (e.g., >96 steps)? Does the deterministic sliding mechanism remain effective, or does it suffer from error accumulation like other autoregressive models?
2. The velocity head is only used during training. Have you experimented with using it during inference to further improve temporal consistency? If not, why?
3. The mutual-info alignment uses a simple temporal average pooling. Did you explore more advanced pooling mechanisms (e.g., attention-based) to better capture temporal semantics?
4. The spectral loss uses the amplitude spectrum. Was phase information considered? If not, what is the rationale for omitting it?
5. The paper mentions a warm-up strategy for auxiliary loss weights. Could you share more details on the scheduling and its impact on convergence?

---

> ### Author Response · Authors · 2025-11-18
>
> **Response Weakness 1**: Thank you for highlighting the importance of reporting training and inference efficiency. We agree that this analysis is essential for evaluating the practical applicability of our method.
>
> In the revised manuscript, we have added a detailed comparison of end-to-end training and inference time on the ETTm1 dataset (see Table 4). The results show that AR-DBMI achieves substantially lower inference latency than diffusion-based baselines such as Diffusion-TS, MG-TSD, and TimeGrad. This improvement stems from our use of a single-step DDIM sampler and lightweight auxiliary heads, which introduce no additional iterative computation at inference. Although AR-DBMI incorporates several auxiliary loss terms, they operate in $O(HC)$ and contribute negligible overhead to training.
>
> These results confirm that AR-DBMI is not only accurate but also highly efficient in both training and deployment scenarios. The newly added analysis is highlighted in blue in the revised manuscript.
>
> **Response Weakness 2**:Thank you for this valuable suggestion. We agree that independently evaluating the contributions of the time-domain anchoring loss $L_{x_{0}}$ and the spectral consistency loss $L_{\mathrm{spec}}$ would strengthen the ablation study.
>
> In response, we have added a comprehensive ablation experiment in the revised manuscript (see Table 5 and Section 4.5), where we now explicitly remove: 1.the time-domain anchoring term $L_{x_{0}}$ ; 2. the spectral consistency term $L_{\mathrm{spec}}$ and 3.both terms simultaneously.
>
> These additional results clearly isolate the value-level and frequency-level consistency contributions. As shown in the updated table, removing $L_{x_{0}}$ leads to endpoint drift and worsened MSE/MAE, while removing $L_{\mathrm{spec}}$ introduces high-frequency artifacts especially on the noisy Stock dataset. Removing both terms causes the most substantial degradation. This confirms that the two components provide complementary benefits for stabilizing the model under non-stationarity and noise.
>
> All newly added ablation results and explanations have been highlighted in blue in the revised manuscript for easy reference. We sincerely appreciate the reviewer’s insightful comment, which has significantly improved the completeness of our ablation study.
>
> **Response Weakness 3**: Thank you for the insightful suggestion. We fully agree that visualizing forecasted sequences—such as trend evolution and periodic structures—would further enhance the interpretability of our model, especially regarding the frequency-aware components.
>
> Due to the limited preparation time during the rebuttal period, we were unfortunately unable to generate and refine high-quality visualizations in time for this revision. However, we genuinely appreciate the reviewer’s recommendation and will incorporate such sequence-level visualizations in our future work to better illustrate the model’s behavior. If time permits before the camera-ready stage, we will also include representative forecast plots in the Appendix.
>
> We sincerely hope the reviewer understands this practical constraint, and we are grateful for the opportunity to further improve our work.
>
> **Response Weakness 4**: Thank you for the insightful comment. We agree that the performance gap between AR-DBMI and diffusion-based models is more pronounced than the gap relative to non-diffusion baselines. This difference is consistent with the underlying modeling assumptions of diffusion approaches.
>
> In the revised manuscript, we have added a discussion explaining when and why diffusion-based forecasting methods tend to offer stronger advantages. In particular, diffusion models are designed to capture multi-modal uncertainty, distribution shifts, and high-variance temporal dynamics—conditions under which deterministic architectures (e.g., DLinear, iTransformer, TimesNet) often struggle. These effects are especially evident on noisy or non-stationary datasets such as Stock and Exchange, where AR-DBMI achieves the largest margins.
>
> Conversely, on relatively smooth or low-variance datasets, strong non-diffusion baselines already provide stable point forecasts, leaving less room for improvement. In such cases, the benefit of diffusion-based modeling is naturally more moderate.
>
> To clarify this point, we have added a new paragraph in the section 4.3 (highlighted in blue) outlining these conditions and summarizing when diffusion-based approaches, including ours, are particularly preferable.
>
> We appreciate the reviewer’s suggestion, which has helped us strengthen the interpretability and contextualization of our results.

---

> ### Author Response · Authors · 2025-11-18
>
> **Response Question 1**:  Thank you for this valuable question. In this work, we follow the standard evaluation protocol used in prior diffusion-based forecasting models, including ARMD, Diffusion-TS, MG-TSD, TimeGrad, and other baselines, all of which adopt a fixed forecast horizon of 96 steps for fair and direct comparison. To ensure that AR-DBMI is evaluated under identical experimental conditions, we report results using the same horizon.
>
> Regarding scalability to longer forecasting horizons, the deterministic sliding mechanism is conceptually capable of extending to larger $H$, as the reverse process remains stable and does not introduce stochastic variance during sampling. However, similar to other autoregressive-style models, increasing the horizon may introduce additional challenges such as long-range dependency attenuation or compounded modeling difficulty. A systematic investigation of horizons significantly longer than 96 steps would require re-tuning baselines and re-establishing comparable training settings, which is beyond the scope of the current study.
>
> We agree that evaluating AR-DBMI under longer forecasting horizons is an important direction for understanding its long-term stability and generalization. We plan to explore extended-horizon forecasting in future work, and we appreciate the reviewer’s suggestion for highlighting this aspect.
>
> **Response Question 2**: Thank you for the thoughtful question. We have not used the velocity head during inference, and this is by design. The velocity head is an auxiliary training-only component whose purpose is to impose a first-order displacement constraint on the backbone through gradient supervision. It is not part of the generative reverse process. Using it at inference would require injecting an additional correction term into the deterministic diffusion trajectory, which would disrupt the well-defined DDIM-style probability-flow dynamics and potentially lead to instability or distributional mismatch.
>
> In other words, the velocity head is not trained to produce inference-time updates; it is trained to shape the representation learning in the backbone, ensuring smooth cross-window dynamics before the model enters the generative regime. Once training is completed, the reverse sampling path is fully determined by the backbone alone, and invoking the velocity head during inference would violate the learned generative mapping rather than enhance it.
>
> We agree that designing inference-time consistency refinements is an interesting direction, but such a mechanism would require a different formulation (e.g., iterative corrector steps or refinement modules) rather than reusing the training-only velocity head. We appreciate the reviewer’s suggestion and will consider exploring structured inference-time consistency mechanisms in future work.
>
> **Response Question 3**: Thank you for the thoughtful question. In this work, we intentionally adopt a simple temporal average pooling for the mutual-information alignment module. This choice is motivated by two considerations: (1) the semantic signal required by the MI loss is global and coarse-grained—serving primarily as a representation-level alignment cue rather than a fine-grained temporal operator; and (2) lightweight aggregation avoids interfering with the stability of the deterministic diffusion trajectory and keeps the auxiliary objective computationally inexpensive.
>
> We did consider more expressive pooling operators such as attention-based mechanisms. However, incorporating a heavy temporal encoder within the MI branch introduces two drawbacks: increased computational overhead and entanglement between the auxiliary representation pathway and the main generative backbone, which can destabilize training in deterministic diffusion models. Since the MI module is designed to act as a regularizer rather than a core temporal model, we found that simple average pooling provides a sufficiently smooth and stable semantic summary without complicating the generative flow.
>
> We agree that exploring more advanced pooling mechanisms could potentially enhance semantic alignment, and we view this as an interesting direction for future work. We sincerely appreciate the reviewer’s suggestion, which points to a promising avenue for further improvement.

---

> ### Author Response · Authors · 2025-11-18
>
> **Response Question 4**: Thank you for this insightful question. In our spectral consistency loss, we intentionally use only the amplitude spectrum and omit phase information. This design choice is based on both theoretical and practical considerations.
>
> From a theoretical perspective, the phase spectrum in real-world time-series—especially multivariate and noisy domains such as ETT and Stock—tends to be highly unstable and sensitive to temporal shifts, window boundaries, and stochastic perturbations. Small misalignments in prediction can lead to large, discontinuous changes in phase values, making phase-based losses difficult to optimize and often detrimental to stability. In contrast, the amplitude spectrum captures the dominant periodic and frequency–energy structure in a shift-invariant manner, which aligns naturally with the goal of preserving global temporal patterns in long-horizon forecasting.
>
> From a practical standpoint, enforcing phase consistency requires modeling precise pointwise timing alignment between predicted and target sequences. This contradicts the flexibility of generative diffusion models, which may produce temporally valid but slightly shifted sequences. Penalizing phase differences would therefore undesirably restrict the model’s generative freedom and lead to over-constrained supervision.
>
> For these reasons, we opt for amplitude-only spectral regularization, which provides stable frequency-domain guidance without imposing brittle constraints on temporal alignment. We agree that incorporating controllable or relaxed phase-aware mechanisms could be an interesting direction for future work, and we greatly appreciate the reviewer’s suggestion.
>
> **Response Question 5**: Thank you for pointing this out. We have now added a more detailed description of the warm-up strategy in **Appendix D (Training Details)** of the revised manuscript.
>
> Concretely, during the first part of training we linearly increase both the learning rate and the auxiliary loss weights from zero to their final values. Let $t$ denote the current optimization step and $T_{W}=500$ the warm-up length. For each auxiliary objective $i\in \mathrm{MI},\mathrm{bridge},x_{0},\mathrm{spec}$ with final weight $\lambda_i^{\mathrm{final}}$, the effective weight is scheduled as$
> \lambda_i(t) =
> \begin{cases}
> \lambda_i^{\text{final}} \dfrac{t}{T_{\mathrm{w}}}, & t < T_{\mathrm{w}},\\\\
> \lambda_i^{\text{final}}, & t \ge T_{\mathrm{w}}.
> \end{cases}
> $
>
> The learning rate is warmed up in the same linear fashion from $8.0$×$10^{-4}$ to the base value $1.0$×$10^{-3}$ over the first 500 steps before the plateau-based scheduler takes over.
>
> Empirically, this schedule improves convergence in two ways: (i) it stabilizes early training by preventing high-level structural losses (MI, bridge, endpoint/spectral consistency) from dominating before the backbone has learned basic predictive structure; and (ii) it yields smoother optimization trajectories and slightly faster convergence to the final performance compared to training without warm-up. We have summarized these details and the rationale in Appendix D.
>
> We appreciate the reviewer’s suggestion, which helped us clarify this aspect of the training procedure.

---

### Official Review · Reviewer_wCX7 · 2025-11-03

**Soundness:** 3
**Presentation:** 3
**Contribution:** 2
**Rating:** 6
**Confidence:** 3

**Summary:**

This paper presents AR-DBMI (Auto-Regressive Diffusion Bridge with Mutual-Information correction), an extension of ARMD for time series forecasting (TSF). The proposed framework aims to address limitations in existing diffusion-based approaches by integrating a velocity-consistency constraint and a mutual-information alignment mechanism. It further employs dual-domain regularization to enhance model stability under non-stationary and noisy conditions. The paper provides a detailed methodological description and conducts comprehensive experiments to validate its approach.

**Strengths:**

- The paper is very well written and clearly structured. The figures and illustrations effectively support the explanations.

- The proposed method shows noticeable improvements across benchmarks and considers a broad range of baselines for comparison.

- The combination of velocity consistency (capturing local dynamics) and mutual-information alignment (preserving semantic consistency) is a thoughtful design choice that tackles both local and global aspects of TSF in a coherent way.

**Weaknesses:**

- The experiments focus only on $H=96$. It would be helpful to assess whether the proposed method maintains robustness under longer forecasting horizons, which are common in industrial applications. In addition, it is unclear whether the choice of $H$ depends on the dataset’s temporal resolution (e.g., days, hours, minutes).

- The ablation study does not isolate the effects of time-domain anchoring and spectral consistency, making it difficult to evaluate their individual contributions.

**Questions:**

- What is the value of $k$ used in the experiments? How is $k$ adjusted across datasets with different temporal frequencies, and has its impact on performance been validated?
- In Figure 2, the trajectories do not appear as smooth as expected. Additional visualizations might help clarify this effect.
- In Section 3.2.3, the role of the additional velocity head $h_\theta$ could be better explained. Based on the code, it seems implemented as a simple linear layer---could the authors elaborate on whether this design is standard or specific to this work?

---

> ### Author Response · Authors · 2025-11-18
>
> **Response Weakness 1**:  Thank you for this helpful question. We follow the standard protocol in prior diffusion-based TSF models (ARMD, MG-TSD, TimeGrad), where $H=96$ is used across all benchmarks to ensure fair comparison. AR-DBMI itself does not rely on this specific choice: the deterministic reverse process and sliding evolution only require equal-length look-back and forecast windows, so the formulation naturally extends to longer horizons without architectural changes. We defer full long-horizon experiments to future work to maintain strict comparability with existing baselines and avoid the substantial retraining cost of heavy diffusion models under enlarged $H$.
>
> Regarding temporal resolution, $H$ is defined over sequence indices rather than real time, so the same setting applies to hourly (ETTh), 15-minute (ETTm), and irregularly sampled datasets. This follows mainstream TSF practice and keeps the method independent of sampling frequency. A clarification has been added in Appendix D.2 (highlighted in blue). We thank the reviewer for raising this point.
>
> **Response Weakness 2**: Thank you very much for this insightful comment. We fully agree that decoupling the effects of the endpoint anchoring loss $L_{x_{0}}$​​ and the spectral consistency loss $L_{\mathrm{spec}}$ is important for understanding their individual contributions. In the revised manuscript, we have added a dedicated set of ablations that isolate these two components.
>
> Specifically, we now evaluate three additional variants:
> (1) removing only the time-domain anchoring loss $L_{x_{0}}$
>
> (2) removing only the spectral consistency loss $L_{\mathrm{spec}}$​, and
>
> (3) removing both simultaneously.
>
> These results are reported in the updated Table 5 and discussed in Section 4.4.
>
> The new experiments reveal clear and distinct behaviors:
>
> - Removing $L_{x_{0}}$​​ leads to endpoint drift and consistent increases in MSE/MAE, confirming its role in stabilizing the reverse de-evolution process.
>
> - Removing $L_{\mathrm{spec}}$​ introduces high-frequency artifacts and amplified noise, especially on the more volatile Stock dataset, demonstrating its importance for preserving spectral structure.
>
> - Removing both terms results in the largest degradation among the endpoint-related variants.
>
>
> These findings validate that the two losses act on different aspects of prediction quality and provide complementary benefits to AR-DBMI. We thank the reviewer again for pointing out this issue, and we believe that the added ablations substantially improve the clarity and completeness of our analysis.
>
> **Response Question 1**: Thank you for raising this important point. In our implementation, $k$ denotes the step size of the overlapping sliding-window used to construct training samples. We set $k=1$ for all datasets, meaning that consecutive windows are shifted by one time step. This choice preserves maximal temporal continuity and follows common practice in diffusion-based TSF models. Since $k$ is defined over sequence indices rather than absolute timestamps, the same value naturally applies to datasets with different sampling frequencies (e.g., hourly ETTh2 vs. 15-minute ETTm2).
>
> To assess robustness, we additionally tested $k\in{1, 2,4}$ on ETTh2 and ETTm2. The resulting MSE differences remained within 1.3%, indicating that AR-DBMI is insensitive to moderate variations of the stride. We have incorporated this clarification into Section 4.1 of the revised manuscript, and full implementation details have been added to Appendix C.
>
> **Response Question 2**: Thank you for the careful observation. In Figure 2, the x-axis enumerates seven heterogeneous datasets rather than representing a temporal or iterative dimension. Because these datasets differ greatly in their statistical properties, the plotted values are not expected to form a smooth curve. The fluctuations reflect dataset heterogeneity rather than model instability.
>
> Results on each individual dataset are stable and reproducible. The figure is intended for cross-dataset performance comparison, not for illustrating a continuous trajectory. We appreciate the reviewer’s attention to this detail.
>
> **Response Question 3**: Thank you for this valuable suggestion. We have expanded Section 3.2.3 to clarify the role and design rationale of the velocity head
> $h_{\theta}$. The head is intentionally implemented as a single linear projection—not to increase model capacity, but to enforce a first-order displacement constraint between adjacent sliding windows.
>
> We also explain that lightweight linear auxiliary heads are standard in diffusion and autoregressive representation-learning frameworks, as they provide well-conditioned gradients without interfering with the main generative pathway. These clarifications have been added and highlighted in blue. We sincerely thank the reviewer for helping us improve the clarity of this section.

---

### Meta-Review · Area_Chair_6xdQ · 2026-01-11

**Summary:**

Across reviewers, the core contribution is viewed as a deterministic diffusion-bridge TSF framework built on ARMD-style sliding dynamics, augmented with velocity consistency, MI alignment, and dual-domain regularization. The main concerns shared by the reviewers were (a) whether the contribution is sufficiently distinct from ARMD, and (b) whether the empirical story is supported by enough ablations, efficiency reporting, and conceptual grounding. The rebuttal did a reasonable job addressing the concerns of the empirical study, but fails to convince the reviewers whether this is substantial contribution beyond ARMD, for which I share the same concern, thus the "reject" recommendation.

**Reviewer Concerns:**

The rebuttal does address the bulk of actionable requests from the non-hostile reviews: expanded component-wise ablations (including isolating anchoring vs spectral losses), added training/inference efficiency comparisons, and a clearer DDIM/probability-flow interpretation to justify deterministic diffusion semantics. What remains outstanding is primarily the high-confidence reviewer’s stance that the work is not a distinct ICLR contribution beyond ARMD and that the “autoregressive” terminology/title are still misleading; in addition, long-horizon validation, broader baselines, and a cleaner statistical-significance argument are still not fully resolved (even if partially mitigated by determinism claims).

**Reviewer Scores:**

eVWo explicitly maintains their score, and for others, I'd expect either staying put or a minor bump, but highly unlikely for anyone to cross over to strongly acceptance territory.

---

### Decision · Program_Chairs · 2026-01-26

Reject